

# Indicators of Global Climate Change 2024: annual update of key indicators of the state of the climate system and human influence

Piers M. Forster [1], Chris Smith [2,3], Tristram Walsh [4], William F. Lamb [1, 5], Robin Lamboll [6], Christophe Cassou [7, 8], Mathias Hauser [9], Zeke Hausfather [10, 11], June-Yi Lee [12, 13], Matthew D. Palmer [14, 15], Karina von Schuckmann [16], Aimée B.A. Slangen [17], Sophie Szopa [18], Blair Trewin [19], Jeongeun Yun [12], Nathan P. Gillett [20], Stuart Jenkins [4], H. Damon Matthews [21], Krishnan Raghavan [22], Aurélien Ribes [23], Joeri Rogelj [3, 6, 24], Debbie Rosen [1], Xuebin Zhang [25], Myles Allen [4, 26], Lara Aleluia Reis [27, 28], Robbie M. Andrew [29], Richard A. Betts [14, 30], Alex Borger [31], Jiddu A. Broersma [31], Samantha N. Burgess [32], Lijing Cheng [33], Pierre Friedlingstein [7, 30], Catia M. Domingues [34, 35], Marco Gambarini [27, 28], Thomas Gasser [3], Johannes Gütschow [36], Masayoshi Ishii [37], Christopher Kadow [38], John Kennedy [39], Rachel E. Killick [14], Paul B. Krummel [40], Aurélien Liné [8, 16, 41], Didier P. Monselesan [42], Colin Morice [14], Jens Mühle [43], Vaishali Naik [44], Glen P. Peters [29], Anna Pirani [45], Julia Pongratz [46, 47], Jan. C. Minx [1, 5], Matthew Rigby [48], Robert Rohde [10], Abhishek Savita [49, 50], Sonia I. Seneviratne [9], Peter Thorne [51], Christopher Wells [1], Luke M. Western [48], Guido R. van der Werf [52], Susan E. Wijffels [42, 53], Valérie Masson-Delmotte [18], Panmao Zhai [54]

1. Priestley Centre for Climate Futures, University of Leeds, Leeds, LS2 9JT, UK
2. Department of Water and Climate, Vrije Universiteit Brussel, Brussels, Belgium
3. International Institute for Applied Systems Analysis (IIASA), Vienna, Austria
4. Environmental Change Institute, University of Oxford, Oxford, UK
5. Potsdam Institute for Climate Impact Research (PIK), Member of the Leibniz Association, Potsdam, Germany
6. Centre for Environmental Policy, Imperial College London, London, UK
7. Laboratoire de Météorologie Dynamique/Institut Pierre-Simon Laplace, CNRS, Ecole Normale Supérieure/Université PSL, Paris, France
8. CECI, Université de Toulouse, CERFACS, CNRS, Toulouse, France
9. Institute for Atmospheric and Climate Science, Department of Environmental Systems Science, ETH Zurich, Zurich, Switzerland
10. Berkeley Earth, Berkeley, CA, USA
11. Stripe Inc., South San Francisco, CA, USA
12. Research Center for Climate Sciences, Pusan National University, Busan, Republic of Korea
13. Center for Climate Physics, Institute for Basic Science, Busan, Republic of Korea
14. Met Office Hadley Centre, Exeter, UK
15. School of Earth Sciences, University of Bristol, Bristol, UK
16. Mercator Ocean international, Toulouse, France



17. NIOZ Royal Netherlands Institute for Sea Research, Department of Estuarine and Delta Systems, Yerseke, the
Netherlands
18. Institut Pierre Simon Laplace, Laboratoire des Sciences du Climat et de l'Environnement (UMR 8212 CEA-CNRS-
UVSQ), Université Paris-Saclay, Gif-sur-Yvette, France
19. Bureau of Meteorology, Melbourne, Australia
20. Canadian Centre for Climate Modelling and Analysis, Environment and Climate Change Canada, Victoria, BC,
Canada
21. Concordia University, Montreal, Canada
22. Indian Institute of Tropical Meteorology, Pune, India
23. CNRM, Université de Toulouse, Météo France, CNRS, Toulouse, France
24. Grantham Institute for Climate Change and Environment, Imperial College London, United Kingdom
25. Pacific Climate Impacts Consortium, University of Victoria, Victoria, Canada
26. Atmospheric, Oceanic and Planetary Physics, Department of Physics, University of Oxford, UK
27. CMCC Foundation, Euro-Mediterranean Center on Climate Change, Lecce, Italy
28. RFF-CMCC, European Institute on Economics and the Environment, Milan, Italy
29. CICERO Center for International Climate Research, Oslo, Norway
30. Global Systems Institute, Science and Economy, University of Exeter, Exeter, UK
31. Climate Change Tracker, Data for Action Foundation, Amsterdam, the Netherlands
32. European Centre for Medium-Range Weather Forecasts, ECMWF, Reading, United Kingdom
33. State Key Laboratory of Earth System Numerical Modeling and Application, Institute of Atmospheric Physics,
Chinese Academy of Sciences, Beijing, China
34. Marine Physics and Ocean Climate, National Oceanography Centre, Southampton, UK
35. Environmental Business Unit, CSIRO, Hobart, Australia
36. Climate Resource, Melbourne, Australia
37. Meteorological Research Institute, Tsukuba, Japan
38. German Climate Computing Center, Hamburg, Germany (DKRZ)
39. No affiliation, Verdun, France
40. CSIRO Environment, Aspendale, Australia
41. Institut de Mécanique des Fluides de Toulouse, Université de Toulouse, INP, CNRS, Toulouse, France
42. CSIRO, Environment Research Unit, Climate Intelligence, Climate variability and hazards, Hobart, Tasmania,
Australia
43. Scripps Institution of Oceanography, University of California San Diego, La Jolla, CA, USA
44. NOAA Geophysical Fluid Dynamics Laboratory, Princeton, NJ, USA
45. Euro-Mediterranean Center on Climate Change (CMCC), Venice, Italy; Università Cà Foscari,Venice, Italy
46. Ludwig-Maximilians-Universität München, München, Germany
47. Max Planck Institute for Meteorology, Hamburg, Germany



48. School of Chemistry, University of Bristol, Bristol, United Kingdom
49. Centre for Atmospheric Sciences, Indian Institute of Technology Delhi, Delhi, India
50. Department of Atmospheric Sciences, Rosenstiel School of Marine, Atmospheric and Earth Science, Miami, USA
51. ICARUS Climate Research Centre, Maynooth University, Maynooth, Ireland
52. Wageningen University and Research, Wageningen, The Netherlands
53. Physical Oceanography, Woods Hole Oceanographic Institution, Woods Hole, Massachusetts, USA
54. Chinese Academy of Meteorological Sciences, Beijing, China

*Correspondence to*: Piers. M. Forster (p.m.forster@leeds.ac.uk)

**Abstract.**
In a rapidly changing climate, evidence-based decision-making benefits from up-to-date and timely information. Here
we compile monitoring datasets (published here, https://doi.org/10.5281/zenodo.15327155 Smith et al., 2025a) to
produce updated estimates for key indicators of the state of the climate system: net emissions of greenhouse gases and
short-lived climate forcers, greenhouse gas concentrations, radiative forcing, the Earth's energy imbalance, surface
temperature changes, warming attributed to human activities, the remaining carbon budget, and estimates of global
temperature extremes. This year, we additionally include indicators for sea-level rise and land precipitation change.
We follow methods as closely as possible to those used in the IPCC Sixth Assessment Report (AR6) Working Group
One (WGI) report.

The indicators show that human activities are increasing the Earth's energy imbalance and driving faster sea-level rise
compared to the AR6 assessment. For the 2015–2024 decade average, observed warming relative to 1850-1900 was
1.24 [1.11 to 1.35] °C, of which 1.23 [1.0 to 1.5] °C was human-induced. The 2024 observed record in global surface
temperature (1.52 °C best estimate) is well above the best estimate of human-caused warming (1.36°C). However, the
2024 observed warming can still be regarded as a typical year, considering the human induced warming level and the
state of internal variability associated with the phase of El Niño and Atlantic variability. Human-induced warming has
been increasing at a rate that is unprecedented in the instrumental record, reaching 0.27 [0.2 - 0.4] °C per decade over
2015-2024. This high rate of warming is caused by a combination of greenhouse gas emissions being at an all-time
high of $53.6 \pm 5.2$ GtCO$_2$e per year over the last decade (2014-2023), as well as reductions in the strength of aerosol
cooling. Despite this, there is evidence that the rate of increase in CO$_2$ emissions over the last decade has slowed
compared to the 2000s, and depending on societal choices, a continued series of these annual updates over the critical
2020s decade could track decreases or increases in the rate of the climatic changes presented here.



## 1 Introduction

IPCC AR6 provided an assessment of human influence on key indicators of the state of climate grounded in available data at the time of publication. The preparation for the next IPCC report, the Seventh Assessment Report (AR7), has started and the assessment is due in around 5 years. Given the speed of recent change, and the need for updated climate knowledge to inform evidence-based decision-making, the Indicators of Global Climate Change (IGCC) was initiated to provide policymakers with annual updates of the latest scientific understanding on the state of selected critical indicators of the climate system and where possible of the quantified human influence upon these.

This third annual update follows broadly the format of last year (Forster et al., 2024), focusing on indicators related to heating of the climate system, building from greenhouse gas emissions towards estimates of human-induced warming and the remaining carbon budget. Fig. 1 presents an overview of the aspects assessed and their interlinkages from cause (emissions) through effect (changes in physical indicators) to Climatic Impact-Drivers. It also provides a visual roadmap as to the structure of remaining sections in this paper to guide the reader.


**Figure 1 The flow chart of data production from emissions to human induced warming, the remaining carbon budget, and changes to Climatic Impact-Drivers, illustrating both the rationale and workflow within the paper production.**

The update is based on methodologies assessed by the IPCC Sixth Assessment Report (AR6) of the physical science basis of climate change (Working Group One (WGI) report; IPCC, 2021a) as well as Chap. 2 of the WGIII report



(Dhakal et al., 2022) and is aligned with the efforts initiated in AR6 to implement FAIR (Findable, Accessible,
Interoperable, Reusable) principles for reproducibility and reusability (Pirani et al., 2022; Iturbide et al., 2022). IPCC
reports make a much wider assessment of the science and methodologies – we do not attempt to reproduce the
comprehensive nature of these IPCC assessments here. We also do not consider adopting fundamentally different
approaches to AR6. Rather, our aim is to rigorously track both climate system change and evolving methodological
improvements between IPCC report cycles, thereby increasing transparency and consistency in between successive
reports.

The update is organised as follows: greenhouse gas (GHG) emissions (Sect. 2), greenhouse gas concentrations (Sect.
3) and emissions of short-lived climate forcers (Sect. 4) are used to develop updated estimates of effective radiative
forcing (Sect. 5). The Earth energy imbalance (Sect. 6) and observations of global surface temperature change (Sect.
7) are key global indicators of a warming world. The contributions to global surface temperature change from human
and natural influences are formally attributed in Sect. 8, which tracks the level and rate of human-induced warming.
Sect. 9 updates the remaining carbon budget for policy-relevant temperature thresholds. Sect. 10 gives an example of
global-scale indicators associated with climate extremes of maximum land surface temperatures and Sect. 11 shows
land-surface precipitation trends traceable to AR6, a new addition to this year's update. Sect. 12 presents updated
estimates of global mean sea-level rise, also a new addition. Code and data availability are given in Sect. 13, and
conclusions are presented in Sect. 14. Data are available at https://doi.org/10.5281/zenodo.15327155 (Smith et al.,
2025a).

**2 Greenhouse gas emissions**
Historic GHG emissions from human activity were assessed in both AR6 WGI and WGIII. Chapter 5 of WGI assessed
$CO_2$ and $CH_4$ emissions in the context of the carbon cycle (Canadell et al., 2021). Chapter 2 of WGIII, published one
year later (Dhakal et al., 2022), assessed the sectoral sources of emissions and gave the most up-to-date understanding
of the current level of emissions. This section bases its methods and data on those employed in this WGIII chapter.
**2.1 Methods of estimating greenhouse gas emissions changes**
Like in AR6 WGIII, net GHG emissions in this paper refer to releases of GHGs from anthropogenic sources minus
removals by anthropogenic sinks, for the set of GHGs outlined in the United Nations Framework Convention on
Climate Change (UNFCCC). These include: $CO_2$ emissions from fossil fuels and industry ($CO_2$-FFI); net $CO_2$
emissions from land use, land-use change and forestry ($CO_2$-LULUCF); $CH_4$ emissions; $N_2O$ emissions; and
fluorinated gas (F-gas) emissions comprising hydrofluorocarbons (HFCs), perfluorocarbons (PFCs), sulphur
hexafluoride ($SF_6$) and nitrogen trifluoride ($NF_3$) - hereafter the "UNFCCC F-gases".



Despite an extensive literature on GHG emissions, there remains important differences in reporting conventions and
system boundaries between assessments. These differences relate to three underlying issues: (1) emissions data sets
vary in their coverage of sources and sectors; (2) there are different approaches to determining the 'anthropogenic'
component of LULUCF emissions and removals; and (3) the Paris Agreement does not cover all relevant sources of
emissions (Lamb et al., 2025).

Concerning the first issue, there are several possible emissions datasets to draw from, each with varying coverage and
update schedules. Emissions data are gathered by countries and submitted to the UNFCCC in the form of national
inventory reports and common reporting tables. However, these "national inventories" are generally incomplete and
are not kept up to date for all countries. Emissions reporting therefore often relies on "third-party" datasets compiled
by research organisations, including: the Global Carbon Budget (GCB; Friedlingstein et al., 2024); the Emissions
Database for Global Atmospheric Research (EDGAR; Crippa et al., 2023); the Potsdam Real-time Integrated Model
for probabilistic Assessment of emissions Paths (PRIMAP-hist; Gütschow et al., 2016; Gütschow et al., 2025)[1]; the
Community Emissions Data System (CEDS; Hoesly et al., 2018; Hoesly and Smith, 2024); and the Global Fire
Emissions Database (GFED; van der Werf et al., 2017). As detailed below, for various reasons not all these datasets
were employed in this update.

Concerning the second issue, there are varying conventions used to quantify $CO_2$-LULUCF fluxes. These include the
use of bookkeeping models and aggregated national inventory reporting (Pongratz et al., 2021), which differ in terms
of their applied system boundaries and definitions, and in particular how they treat "indirect anthropogenic effects"
such as the influence of increased atmospheric $CO_2$ on vegetation growth. As such, the $CO_2$-LULUCF emissions
estimates generated using bookkeeping models versus national inventories are not directly comparable and differ by
about 7.5GtCO2yr-1 (2013-2022 average), but there are now methods to "translate" between these two approaches
(Friedlingstein et al., 2022; Grassi et al., 2023; Schwingshackl et al., 2022). Assessments also differ with respect to
biomass fire emissions and to what extent components of these are treated as anthropogenic (Lamb et al., 2025).

Finally, two categories of emissions are not directly covered by the Paris Agreement but might be considered
depending on the objectives of an assessment. These include the Ozone Depleting Substances (hereafter the "ODS F-
gases") comprising halons, chlorofluorocarbons (CFCs) and hydrochlorofluorocarbons (HCFCs). The ODS F-gases
were initially controlled under the Montreal Protocol and its amendments and are therefore not included in national
inventories submitted to the UNFCCC, nor in many third-party emissions datasets - in contrast to the UNFCCC F-
gases. Another important omission is the cement carbonation sink. To date this has also been excluded from national

---

[1] PRIMAP is a synthetic dataset that includes two time-series: PRIMAP Hist-TP, which is compiled from other underlying products such as EDGAR; and PRIMAP Hist-CR, which prioritises data from national inventories but gap-fills these where necessary.



reporting under the UNFCCC, but plans for a new chapter covering these removals in the IPCC Task Force on National
Greenhouse Gas Inventory Guidelines indicates a pathway for its eventual inclusion (IPCC, 2025).

The IPCC AR6 WGIII addressed these issues as follows. Total net GHG emissions were calculated as the sum of $CO_2$-
FFI, $CH_4$, $N_2O$ and UNFCCC F-gases from EDGAR (version 6, with a fast-track methodology applied for the final
year of data - 2019), and net $CO_2$-LULUCF emissions from the GCB (the 2020 version; Friedlingstein et al., 2020).
Net $CO_2$-LULUCF emissions followed the GCB convention and were derived from the average of three bookkeeping
models (Hansis et al., 2015; Houghton and Nassikas, 2017; Gasser et al., 2020). "Indirect anthropogenic effects" on
the terrestrial carbon fluxes were therefore excluded from totals (i.e., they were treated as part of the natural land sink).
Further, the GCB methodology (and thus reporting in IPCC AR6 WGIII) includes $CO_2$ emissions from deforestation
and forest degradation fires, but excludes those from wildfires, which are classified as natural even if climate change
affects their intensity and frequency. Similarly, the EDGAR dataset used in AR6 includes some non-$CO_2$ biomass fire
emissions in the agricultural sector, but otherwise excludes those from wildfires. Sources not covered by inventories
or the Paris Agreement (ODS F-gases and cement carbonation) were also excluded. Together these choices ensured
consistency with the Integrated Assessment Model (IAM) benchmarks reported in WGIII and were closely focused
on direct anthropogenic emissions under the UNFCCC, reflecting the importance of human-driven technology and
policy options in shaping the future climate response.

The analysis presented here continues to provide an "WGIII update" estimate that tracks the same system boundary
and compilation of GHGs as in AR6 WGIII, albeit with some differences in the selected data sources. As in previous
years, we use GCB data for $CO_2$-FFI. We also continue to use GCB for $CO_2$-LULUCF, which has now been updated
to use the average of four (rather than three) bookkeeping models (BLUE by Hansis et al., 2015; H&C by Houghton
and Castanho, 2023; OSCAR by Gasser et al., 2020; LUCE by Qin et al., 2024). We use PRIMAP Hist-TP data for
$CH_4$ and $N_2O$, and inversions of atmospheric concentrations tracked by NOAA and AGAGE with best-estimate
lifetimes for UNFCCC F-gas emissions based on analysis in the subsequent Section 3 (Lan et al., 2025; Dutton et al.,
2024; Prinn et al., 2018). We follow the same approach for estimating uncertainties and $CO_2$-equivalent emissions as
in AR6, as described in the Supplement.

In addition to the WGIII update, we provide two further estimates that provide clarity and comparison to other
assessment approaches. This reflects the fact that other decision criteria for tracking emissions are possible. First, in
cases where assessments prioritise calculating the best estimate of fluxes to the atmosphere, it would be important to
include ODS F-gases, cement carbonation and all non-$CO_2$ biomass fire emissions, including those from wildfires.
Indeed, these are included in this article in subsequent assessments of concentration change (including compounds
formed in the atmosphere as ozone), effective radiative forcing, human-induced warming, carbon budgets and climate
impacts, in line with the WGI assessment. We therefore provide an "IPCC update + additional sources and sinks"
estimate that shows the change implied by including these three components in the global total. Second, the IPCC





AR7 report outline foresees the tracking of "inventory-aligned" emissions that are consistent with national reporting.
Full alignment between emissions inventories and WGIII emissions consistent with IAM benchmarks is essential for
an accurate assessment and stocktake of the Nationally Determined Contributions (NDCs) and pathways to net-zero
emissions (Grassi et al., 2021; Gidden et al., 2023; Allen et al., 2025). We therefore provide an "Inventory-aligned"
estimate that follows the inventory approach to accounting for LULUCF emissions, while also integrating the latest
national inventory data from the Common Reporting Tables. The data sources associated with these additional
estimates are detailed in Table S1 in the Supplement.

We expect to see differences between the three estimates, most notably between the "WGIII update" and "Inventory-
aligned" estimates. As discussed above, these differ conceptually in their treatment of the LULUCF sector. However,
national inventory reporting can also differ from third-party datasets in terms of underlying methods: in some
countries, investments into statistical infrastructures have enabled the use of more precise emissions factors in
inventories to estimate fluxes according to local or national conditions, while in others this may not be the case. In
contrast, third-party datasets often use globally consistent emissions factors. Notably, the PRIMAP Hist-CR dataset,
which is here used to represent national inventories, has significantly lower total $CH_4$ emissions relative to other
datasets reported here, as well as the global atmospheric inversion estimates evaluated in this paper. A substantive
body of literature has found that, on average, national inventories tend to underestimate $CH_4$ compared to inversions
(Deng et al., 2022; Tibrewal et al., 2024; Janardanan et al., 2024; Scarpelli et al., 2022).
**2.2 Updated greenhouse gas emissions**
Updated GHG emission estimates following the WGIII assessment are presented in Fig. 2 and Table 1. Total global
GHG emissions were $55.1 \pm 5.1 \, GtCO_2e$ in 2023. Of this total, $CO_2$-FFI contributed $37.8 \pm 3.0 \, GtCO_2$, $CO_2$-LULUCF
contributed $3.6 \pm 2.5 \, GtCO_2$, $CH_4$ contributed $9.2 \pm 2.7 \, GtCO_2e$, $N_2O$ contributed $2.9 \pm 1.7 \, GtCO_2e$ and F-gas
emissions contributed $1.6 \pm 0.5 \, GtCO_2e$.

Initial projections for 2024 indicate that $CO_2$ emissions from fossil fuels and industry increased to $38.2 \pm 3.0$, and $CO_2$
emissions from land-use change increased to $4.2 \pm 2.8 \, GtCO_2$ (Friedlingstein et al., 2024; Deng et al., 2024). The
significant increase in land-use change emissions is connected to high emissions from tropical deforestation and
degradation fires in the aftermath of the El Niño with droughts in South America continuing since 2023. Synchronous
large fires occurred in North America, where the record-breaking Canadian fires of 2023 were followed by another
well above average year in 2024, but are attributable to climate variability and climate change, and not anthropogenic
land-use change (Friedlingstein et al., 2024).

Average annual GHG emissions for the decade 2014–2023 were $53.6 \pm 5.2 \, GtCO_2e$. Average decadal GHG emissions
have increased steadily since the 1970s across all major groups of GHGs, driven primarily by increasing $CO_2$
emissions from fossil fuel and industry but also rising emissions of $CH_4$ and $N_2O$. Emissions of UNFCCC F-gases





have grown more rapidly than other GHG, but from low levels. Both the magnitude and trend of $CO_2$ emissions from
land-use change remain highly uncertain, with the latest data indicating an average net flux between 4–5 $GtCO_2$ $yr^{-1}$
for the past few decades.

The fossil fuel share of global GHG emissions was approximately 70% in 2023 (GWP100 weighted), based on the
EDGAR v9 dataset (Crippa et al., 2023) and net land-use $CO_2$ emissions from the Global Carbon Budget
(Friedlingstein et al. 2024). The remaining share of non-fossil fuel emissions are mostly from land-use change,
agriculture, cement production, waste and F-gas emissions.

Different emissions assessment approaches are shown in Fig. 3. Increasing the scope of the WGIII update to include
ODS F-gases, cement carbonation, and $CH_4$ and $N_2O$ from biomass burning results in emissions of $56.6 \pm 5.2$ GtCO2e
$yr^{-1}$ in 2023, or a total change of $+1.5$ GtCO2e $yr^{-1}$. ODS F-gas emissions have declined substantially since the 1990s
under the Montreal Protocol and its amendments, reaching 1.4 GtCO2e $yr^{-1}$ in 2023, with a stalling rate of reduction
in the past decade. The cement carbonation sink has steadily increased alongside cement production to reach -0.8
GtCO2e $yr^{-1}$ in 2023. Biomass fire emissions have a more variable trend and 2023 was a relatively extreme year at 1
GtCO2e $yr^{-1}$, compared to an average of 0.7 GtCO2e $yr^{-1}$ in the preceding decade.

Emissions according to national inventories were $47.1 \pm 4.7$ GtCO2e $yr^{-1}$ in 2023, or 7.9 GtCO2e $yr^{-1}$ lower than the
WGIII update (Fig. 3). The main reason is due to diverging estimates of net LULUCF emissions, which according to
inventory accounts were on average 7.5 $GtCO_2$ lower over the past decade (2014-2023). Additional differences result
from a lower estimate of Energy, Industrial Process, Agriculture and Waste emissions in inventories (-1.5 GtCO2e
$yr^{-1}$), particularly for $CH_4$ (-0.7 GtCO2e $yr^{-1}$).

Emerging literature, published after AR6 suggests that increases in atmospheric $CH_4$ concentrations may also be
driven by methane emissions from wetland changes resulting from climate change and variability (e.g. Basu et al.,
2022; Hardy et al., 2023; Peng et al., 2022; Nisbet et al., 2023; Zhang et al., 2023). There is also a possible effect from
$CO_2$ fertilisation (Feron et al., 2024; Hu et al., 2023). The latest global methane budget estimates indirect
anthropogenic $CH_4$ fluxes from wetlands and freshwater bodies of approximately 2.4 GtCO2e $yr^{-1}$ (Saunois et al.,
2024). Such emissions are not captured in the WGIII estimate here as they are not a direct emission from human
activity, but rather a feedback induced by a changing climate, yet they will contribute to GHG concentration rise,
forcing and energy budget changes discussed in the next sections. They will become more important to properly
account for in future years. Note that these indirect $CH_4$ emissions are not used to determine the effective radiative
forcing in Sect. 5.



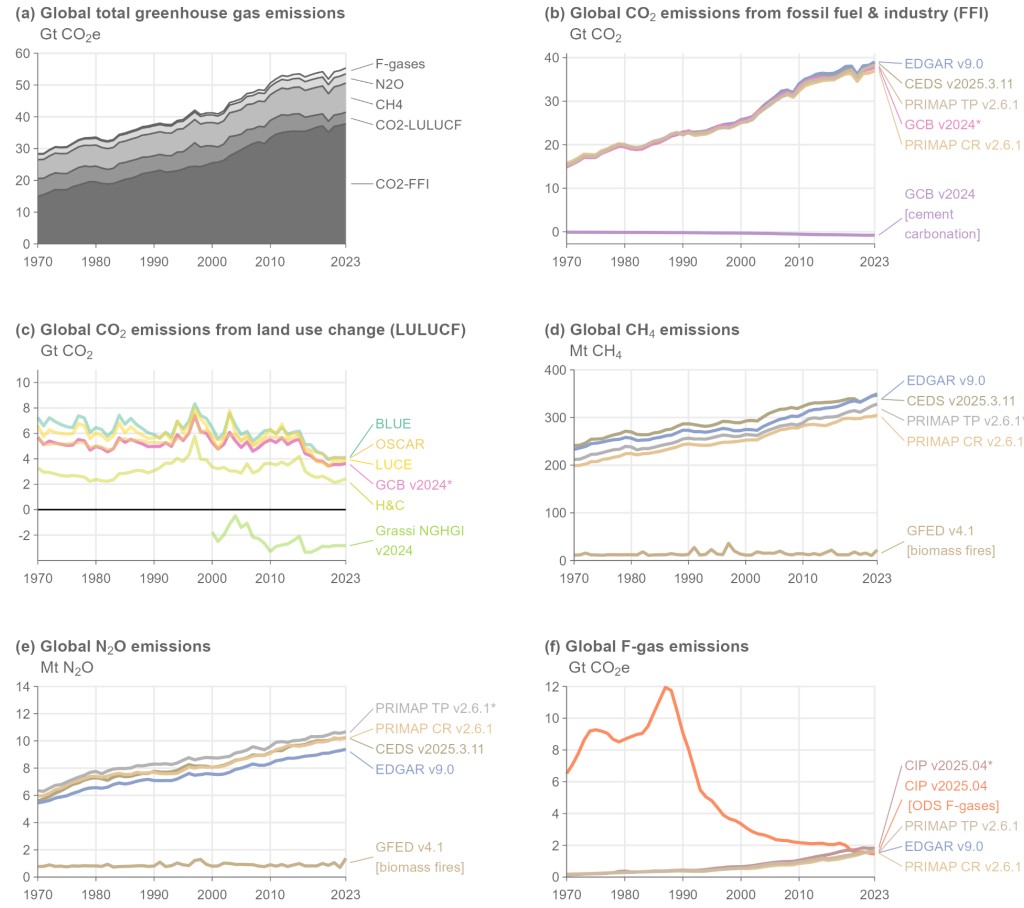

**Figure 2 Annual global anthropogenic GHG emissions by source, 1970–2023. Refer to Sect. 2.1 and Table S1 for a list of datasets. Datasets with an asterisk (*) indicate the sources used to compile global total greenhouse gas emissions following the WGIII assessment in (a). CO₂-equivalent emissions in (a) and (f) are calculated using GWP100 from the AR6 WGI Chap. 7 (Forster et al., 2021). F-gas emissions in (a) comprise only UNFCCC F-gas emissions (see Sect. 2.1 for a list of species). F-gas emissions in (f) refer to UNFCCC F-gases, except for "CIP v2024.04 [ODS F-gases]". Some of the major depicted differences between datasets (e.g. between GCB v2024 and Grassi NGHGI v2024 in panel c) are due to varying system boundaries, rather than underlying uncertainties in activity levels or emissions factors.**



**Table 1 Global anthropogenic greenhouse gas emissions by source and decade following the WGIII assessment. All numbers**
**refer to decadal averages, except for annual estimates in 2023 and 2024. CO₂-equivalent emissions are calculated using**
**GWP100 from AR6 WGI Chap. 7 (Forster et al., 2021). Projections of non-CO₂ GHG emissions in 2024 remain unavailable**
**at the time of publication. Uncertainties are ±8 % for CO₂-FFI, ±70 % for CO₂-LULUCF, ±30 % for CH₄ and F-gases, and**
**±60 % for N₂O, corresponding to a 90 % confidence interval.**

| Units: GtCO₂e | 1970-1979 | 1980-1989 | 1990-1999 | 2000-2009 | 2010-2019 | 2014-2023 | 2023 | 2024 (projection) |
|---|---|---|---|---|---|---|---|---|
| GHG | 30.9±4.5 | 34.6±4.6 | 39.3±5.1 | 45.1±5.1 | 52.9±5.4 | 53.6±5.2 | 55.1±5.1 | |
| CO₂-FFI | 17.3±1.4 | 20.3±1.6 | 23.6±1.9 | 28.9±2.3 | 35.4±2.8 | 36.3±2.9 | 37.8±3.0 | 38.2±3.0 |
| CO₂-LULUCF | 5.2±3.7 | 5.1±3.6 | 5.7±4.0 | 5.2±3.6 | 4.9±3.4 | 4.1±2.9 | 3.6±2.5 | 4.2±2.8 |
| CH₄ | 6.3±1.9 | 6.7±2 | 7.2±2.2 | 7.7±2.3 | 8.4±2.5 | 8.7±2.6 | 9.2±2.7 | |
| N₂O | 1.9±1.1 | 2.2±1.3 | 2.3±1.4 | 2.5±1.5 | 2.7±1.6 | 2.8±1.7 | 2.9±1.7 | |
| UNFCCC F-gases | 0.2±0.01 | 0.4±0.1 | 0.5±0.2 | 0.8±0.3 | 1.4±0.4 | 1.6±0.5 | 1.6±0.5 | |



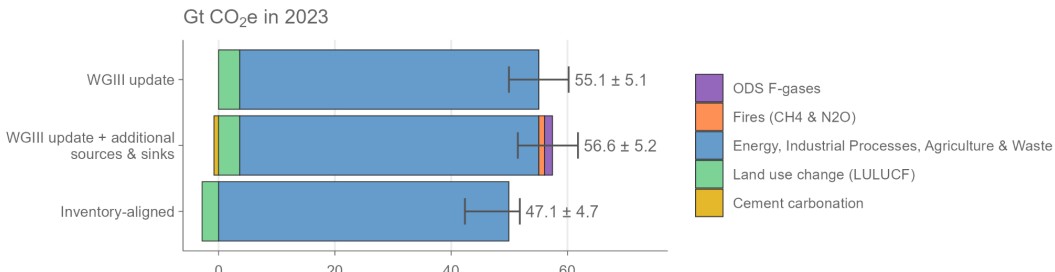


**Figure 3 Annual global anthropogenic greenhouse gas emissions by assessment convention in 2023. Refer to Table 1 for a**
**list of underlying datasets. Differences between conventions are primarily due to differences in system boundaries (Lamb**
**et al., 2025). Uncertainties are ±8 % for CO₂-FFI, ±70 % for CO₂-LULUCF, ±30 % for CH₄ and F-gases, and ±60 % for**
**N₂O, corresponding to a 90 % confidence interval.**





## 3 Well-mixed greenhouse gas concentrations

As in Forster et al. (2024), we report best-estimate global mean concentrations for 52 well-mixed GHGs. These concentrations are updated to 2024. $CO_2$ mixing ratios were taken from the NOAA Global Monitoring Laboratory (GML) and are updated here through 2024 (Lan et al., 2025a). As in Forster et al. (2023, 2024), $CO_2$ is reported on the WMO-$CO_2$-X2019 scale, which differs from the WMO-$CO_2$-X2007 scale used in AR6 with WMO-$CO_2$-X2019 being around 0.18 ppm higher than WMO-$CO_2$-X2007 in recent years. For consistency with WMO-$CO_2$-X2019, the AR6 $CO_2$ concentrations that make up the 1750 to 1978 period in the IGCC dataset (before recent NOAA updates) have been converted to the WMO-$CO_2$-X2019 scale. Other GHG records were compiled from NOAA and AGAGE global networks or extrapolated from literature. An average of NOAA and AGAGE data, updated through 2024, were used for $N_2O$, $CH_4$, CFC-11, CFC-12, $CCl_4$, HCFC-22, HFC-134a, and HFC-125 (Lan et al., 2025; Dutton et al., 2024; Prinn et al., 2018), which, along with $CO_2$, account for over 97% of the ERF from well-mixed GHGs. Several other species also use means from the NOAA and AGAGE networks, where the NOAA data is updated to 2024 from the values given in the BAMS State of the Climate Report (Dunn et al., 2024) and AGAGE data up until 2022 is available; for 2023 and 2024, an offset to the NOAA data was applied which was equal to the mean difference between the NOAA and AGAGE datasets over the recent past. In cases where no updated information is available, global estimates were extrapolated from Vimont et al. (2022), Western et al. (2023, 2024), or other literature and scaled to be consistent with those reported in AR6. Some extrapolations are based on data from the mid-2010s (Droste et al., 2020; Laube et al., 2014; Simmonds et al., 2017; Vollmer et al., 2018), but have an imperceptible effect on the total ERF assessed in Sect. 5, and are included to maintain consistency with AR6. Mixing ratio uncertainties for 2024 are assumed to be like 2019, and we adopt the same uncertainties as assessed in AR6 WGI.

Fig. 4 shows recent GHG concentrations and their changes. Table S2 in the Supplement shows specific updated concentrations for all the GHGs considered. The global surface mean concentrations of $CO_2$, $CH_4$ and $N_2O$ in 2024 were 422.8 [±0.4] parts per million (ppm), 1929.7 [±3.3] parts per billion (ppb) and 337.9 [±0.4] ppb, respectively. Concentrations of all three major GHGs have increased since 2019, with $CO_2$ increasing by 12.7 ppm, $CH_4$ by 63.3 ppb, and $N_2O$ by 5.8 ppb. Increases since 2019 are consistent with those from the CSIRO network (Francey et al., 1999), which are 13.0 ppm, 61.9 ppb, and 6.0 ppb for $CO_2$, $CH_4$, and $N_2O$, respectively. With few exceptions, concentrations of ozone-depleting substances, such as CFC-11 and CFC-12, continue to decline, while those of replacement compounds (HFCs) have increased. HFC-134a, for example, has increased 25% since 2019 from 107.6 to 134.7 parts per trillion (ppt). Aggregated across all gases, PFCs have increased from 109.7 to an estimated 117.4 ppt $CF_4$-eq from 2019 to 2024, HFCs from 237 to 312 ppt HFC-134a-eq, while ozone depleting gases have declined from 1032 to 996 ppt CFC-12-eq. Mixing ratio equivalents are determined by the radiative efficiencies of each GHG from Hodnebrog et al. (2020).

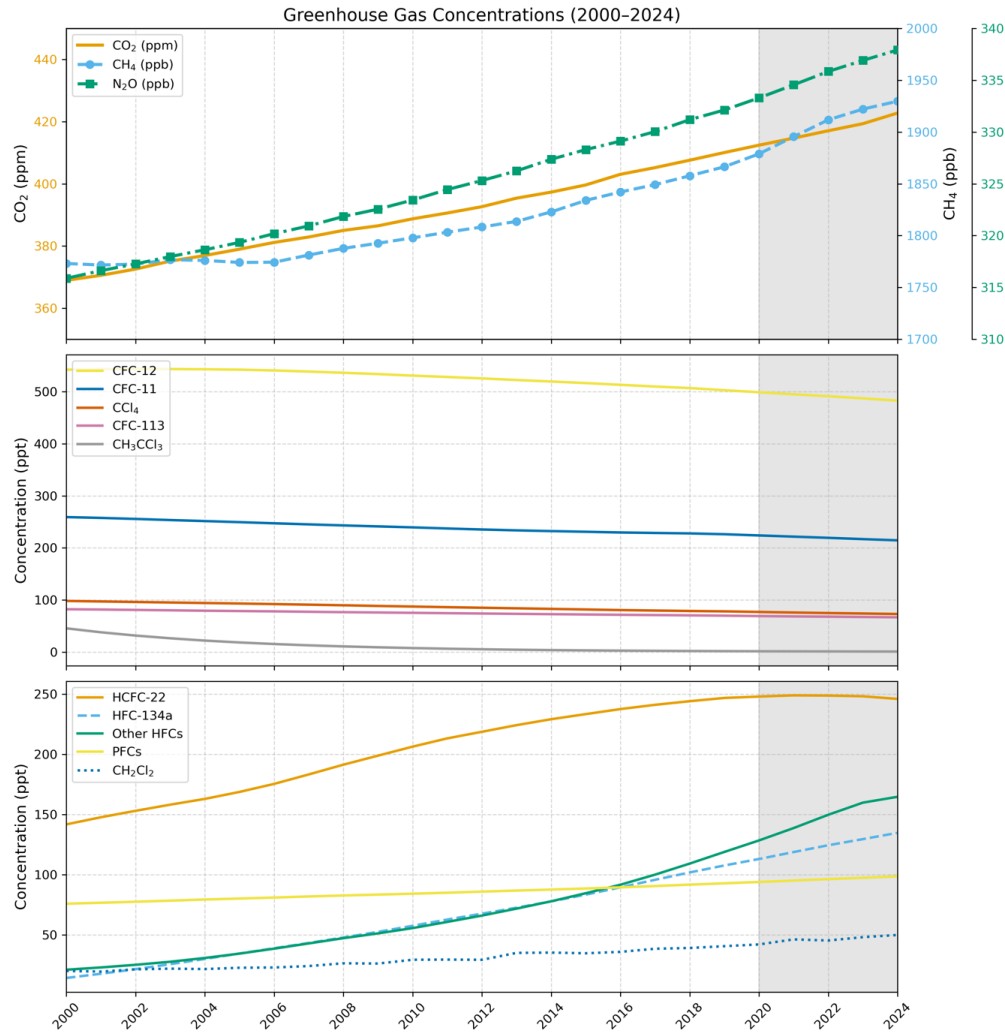


**Figure 4 Atmospheric concentrations of a set of well mixed greenhouse gases over 2000-2024. The grey shaded region**
**represents continuing changes since AR6. Note the different vertical scales.**


Ozone and other non-methane SLCFs are not well-mixed in the atmosphere and are thus discussed separately (in
Section 4). For this reason, the warming impact of ozone, the third most important GHG (in terms of current
contribution to warming) is not included in the contribution of well-mixed GHGs to observed warming, consistently
with AR6. Note that change in methane concentration affects ozone, but this indirect effect is not accounted for in the
estimate of the warming due to the evolution of in well mixed GHG concentrations.





## 4 Non-methane short-lived climate forcers

Chapter 6 of WGI assessed emissions in the context of understanding the climate and air quality impacts of SLCFs (Szopa et al., 2021). Methane is a SLCF but also a well mixed GHG and is discussed in Sections 2 and 3. Trends in SLCFs emissions are spatially heterogeneous (Szopa et al., 2021), with strong shifts in the locations of reductions and increases over the decade 2010–2019 (Hodnebrog et al., 2024). Concentrations of non-methane SLCFs are heterogeneously distributed in the atmosphere and the observation networks are too sparse to report globally averaged concentrations. Typically, a combination of satellite data, where available, and global models and reanalysis are relied upon for producing global-scale distributions. In the case of models, production of near-real time information relies upon the availability of near-real time updates to SLCF emissions which are still challenging. Little information, whether from observations from local monitoring networks, satellite data or from global model reanalysis, is released in near real time.

In addition to GHG emissions, we provide an update of anthropogenic emissions of non-methane SLCFs ($SO_2$, black carbon (BC), organic carbon (OC), NOx, volatile organic compounds (VOCs), CO and $NH_3$). Data are presented in Table 2 and the evolution of SLCF emission estimates from the AR6 to this study is presented in Sect. S4 of the Supplement. Consistency between emission trends and concentrations is considered whenever feasible. HFCs, whatever their lifetimes, were considered in Sect. 2.2.

Sectoral emissions of SLCFs are derived from two sources: CEDS, which was used in the AR6 and in CMIP6 to assess historical evolution of atmospheric composition and that has been updated since then, and the Copernicus Atmosphere Monitoring Service (CAMS). The most recent release of the CEDS anthropogenic emissions dataset (Hoesly et al., 2025) covers the 1750-2023 period (Hoesly et al., 2018; Hoesly and Smith, 2024). Since 2023, CAMS has released regular updates of their global emission dataset (Soulie et al., 2023). For the year 2024, we apply, for each compound, the trend in emission from the CAMS dataset to the 2023 CEDS emission. The CAMS dataset is essentially based on the EDGARv6/v7 emissions as well as on CEDS, so CEDS and CAMS are not entirely independent. The temporal extension is based on evolution of drivers of emissions (energy consumption, production rates) and trends in technologies that affect the emissions factors (e.g. fleet renewal and abatement systems) (Denier van der Gon et al., 2023).

The CAMS v6.2 emission dataset (ECCAD, 2025) indicates a decrease in global anthropogenic emissions of the primary SLCFs (NOx, CO, NMVOCs, $SO_2$, BC and OC) since the COVID hiatus in emissions, except for $NH_3$, whose emissions are steadily increasing. SLCF emissions from biomass burning are taken from GFED (van der Werf et al., 2017) with small fires (GFED4.1s) updated to 2024 (following AR6 WGIII (Dhakal et al., 2022)). Estimates from GFED for 2017 to 2024 are provisional and will be updated with GFED5 in future datasets which will provide substantially higher emissions for most species. The estimate of global carbon emissions due to wildfires in 2024 is slightly lower than in 2023 (both were higher than average fire years). These lower overall



carbon emissions in 2024 hide an increase in $CO_2$ emissions (accompanied by an increase in NOx emissions) but a
decrease in $CH_4$ and CO emissions accompanied by a decrease in carbonaceous aerosols and NMVOC emissions.

The decrease of global NOx emissions, despite very heterogeneous regional trends (Szopa et al., 2021), is confirmed
by global $NO_2$ satellite observations from OMI (tropospheric $NO_2$ column from OMI visualised through the
Giovanni system, Acker and Leptoukh, 2007). The trends in global CO concentration are less clear. Surface data
from MOPITT and AIRS show a slight increase over the last three years. CO does not result solely from CO
emissions but also from VOC including methane oxidation which can explain differences in trends between
emissions and concentrations.

Overall, the trends in emissions were similar (see Supplement Sect. S4) over the 2020-23 period in the most recent
CEDS dataset to our previous estimate (Forster et al., 2024) but with a lower post COVID rebound for NOx and $SO_2$.
Regarding $SO_2$, the CEDS datasets (v2024_04_01 used in Forster et al., 2024 and v2025_03_18 used here) account
for the introduction of strict fuel sulphur controls brought in by the International Maritime Organization in January
2020. Total $SO_2$ emissions in 2019 were 80.9 $TgSO_2$ (Table 2). The $SO_2$ emissions from international shipping
declined by 8.4 $TgSO_2$ from 10.4 $TgSO_2$ in 2019 to 2.0 $TgSO_2$ in 2020, which is close to the expected 8.5 $TgSO_2$
reduction estimated by the International Maritime Organization. This decrease was estimated at 7.4 $TgSO_2$ in the
previous CEDS version used in Forster et al. (2024). More generally, the reduction pace of the global $SO_2$ emission
over the last ten years corresponds to that of the first ten years of the SSP scenarios assuming strong air pollution
control (SSP1 and SSP5).

Using our combined estimate of GFED, and CEDS (with a 2024 extrapolation based on CAMS), emissions of all
SLCFs were reduced in 2022 relative to 2019, but rebounded in 2023 and then slightly decreased in 2024 (relative to
2023) for all compounds except NOx whose increase is partly driven by increased emissions from biomass burning
(Table 2 and Supplement Sect. S4).  2023 was a record year for emissions of organic carbon (driven again by a very
active biomass burning season) and ammonia (driven by a steady background increase in agricultural sources, plus a
contribution from biomass burning). Fires can be worsened by climate change, because of increased fire prone weather
conditions (Burton et al., 2024). Strictly speaking, such fires should be considered as feedbacks and not be included
in anthropogenic forcings. However, we choose to include fires in our tracking, as historical biomass burning
emissions inventories have previously been consistently treated as an anthropogenic forcing (for example in CMIP6),
though this assumption may need to be revisited in the future (see also discussion in Sect. 5). This differs from the
treatment of accounting for $CO_2$ and $CH_4$ emissions at present (Sect. 2.2), where we do not include natural emissions
in the inventories.  As described in Sect. 5, this treatment of all biomass burning emissions as a forcing has implications
for several categories of anthropogenic radiative forcing.



**Table 2 Emissions of the major SLCFs in 1750, 2019, and 2024 from a combination of CEDS and GFED. Emissions of SO₂+SO₄ use SO₂ molecular weights. Emissions of NOₓ use NO₂ molecular weights. VOCs are for the total mass.**

| Compound | SLCF emissions (Tg yr$^{-1}$) | | | | |
|---|---|---|---|---|---|
| | 1750 | 2019 (WGI for ERF estimates) | 2019 (updated) | 2023 (updated) | 2024 (updated) |
| Sulphur dioxide (SO$_2$) + sulfate (SO$_4^{2-}$) | 2.8 | 84.2 | 80.9 | 72.7 | 71.2 |
| Black carbon (BC) | 2.1 | 7.5 | 7.3 | 7.6 | 7.5 |
| Organic carbon (OC) | 15.5 | 34.2 | 33.0 | 41.0 | 36.1 |
| Ammonia (NH$_3$) | 6.6 | 67.6 | 66.3 | 72.7 | 70.6 |
| Oxides of nitrogen (NOₓ) | 19.4 | 141.7 | 133.6 | 128.4 | 130.4 |
| Volatile organic compounds (VOCs) | 60.9 | 217.3 | 204.8 | 224.1 | 212.7 |
| Carbon monoxide (CO) | 348.4 | 853.8 | 816.1 | 896.0 | 845.3 |

Uncertainties associated with these emission estimates are difficult to quantify. From the non-biomass-burning sectors they are estimated to be smallest for SO$_2$ (±14 %), largest for black carbon (BC) (a factor of 2) and intermediate for other species (Smith et al., 2011; Bond et al., 2013; Hoesly et al., 2018). Relative uncertainties are also likely to increase both backwards in time (Hoesly et al., 2018) and again in the most recent years. Future updates of CEDS are expected to include uncertainties (Hoesly et al., 2018).

## 5 Effective radiative forcing (ERF)

ERFs were principally assessed in Chap. 7 of AR6 WGI (Forster et al., 2021), which focussed on assessing ERF from changes in atmospheric concentrations; it also supported estimates of ERF in Chap. 6 that attributed forcing to specific



precursor emissions (Szopa et al., 2021) and generated the time history of ERF shown in AR6 WGI Fig. 2.10 and
discussed in Chap. 2 (Gulev et al., 2021).

The ERF calculation follows the methodology used in AR6 WGI (Smith et al., 2021) as updated by Forster et al.
(2024) and described in the Supplement Sect. S5). One methodological update is incorporated in IGCC 2024 for the
ERF from land use surface reflection and irrigation (Supplement Sect. S5.4).  For each category of forcing, a 100,000-
member probabilistic Monte Carlo ensemble is sampled to span the assessed uncertainty range in each forcing.
Uncertainties account for systematic, structured random and random components. All uncertainties are reported as
5 %–95 % ranges and provided in square brackets. The methods are all detailed in the Supplement, Sect. S5.

The summary results for the anthropogenic constituents of ERF and solar irradiance in 2024 relative to 1750 are shown
in Fig. 5a. In Table 3 these are summarised alongside the equivalent ERFs from AR6 (1750–2019) and last year's
Climate Indicators update (1750-2023). Fig. 5b shows the time evolution of ERF from 1750 to 2024.

**Table 3 Contributions to anthropogenic effective radiative forcing (ERF) for 1750–2024 assessed in this section. Data is for**
**single year estimates unless specified. All values are in watts per square metre (W m$^{-2}$), and 5 %–95 % ranges are in square**
**brackets. As a comparison, the equivalent assessments from AR6 (1750–2019) and last year's Climate Indicators (1750-**
**2023) are shown. Solar ERF is included and unchanged from AR6, based on the most recent solar cycle (2009–2019), thus**
**differing from the single-year estimate in Fig. 5a. Volcanic ERF is excluded due to the sporadic nature of eruptions.**

| Forcer | 1750-2019 [W m$^{-2}$] (AR6) | 1750-2023 [W m$^{-2}$] (Forster et al., 2024) | 1750-2024 [W m$^{-2}$] | Reason for change since last year |
|---|---|---|---|---|
| $CO_2$ | 2.16 [1.90 to 2.41] | 2.28 [2.01 to 2.56] | 2.33 [2.05 to 2.61] | Increases in GHG concentrations resulting from increases in emissions |
| $CH_4$ | 0.54 [0.43 to 0.65] | 0.56 [0.45 to 0.68] | 0.57 [0.45 to 0.68] | |
| $N_2O$ | 0.21 [0.18 to 0.24] | 0.22 [0.19 to 0.26] | 0.23 [0.20 to 0.26] | |
| Halogenated GHGs | 0.41 [0.33 to 0.49] | 0.41 [0.33 to 0.49] | 0.41 [0.33 to 0.49] | |
| Ozone | 0.47 [0.24 to 0.71] | 0.51 [0.25 to 0.76] | 0.50 [0.25 to 0.75] | |
| Stratospheric water vapour | 0.05 [0.00 to 0.10] | 0.05 [0.00 to 0.10] | 0.05 [0.00 to 0.11] | |
| Aerosol-radiation interactions | -0.22 [-0.47 to +0.04] | -0.26 [-0.50 to -0.03] | -0.22 [-0.44 to +0.01] | Decrease in most aerosol and aerosol precursor emissions (Table 2) |
| Aerosol-cloud interactions | -0.84 [-1.45 to -0.25] | -0.91 [-1.80 to -0.27] | -0.85 [-1.65 to -0.25] | |



| Land use (surface albedo changes and effects of irrigation) | -0.20 [-0.30 to -0.10] | -0.20 [-0.31 to -0.10] | -0.19 [-0.30 to -0.05] | Separation of albedo and irrigation components; updated data source and methodology |
|---|---|---|---|---|
| Light-absorbing particles on snow and ice | 0.08 [0.00 to 0.18] | 0.08 [0.00 to 0.17] | 0.08 [0.00 to 0.19] | |
| Contrails and contrail-induced cirrus | 0.06 [0.02 to 0.10] | 0.05 [0.02 to 0.09] | 0.06 [0.02 to 0.10] | |
| Total anthropogenic | 2.72 [1.96 to 3.48] | 2.79 [1.78 to 3.61] | 2.97 [2.05 to 3.76] | Increasing positive GHG forcing and decreasing negative aerosol forcing |
| Solar irradiance | 0.01 [-0.06 to 0.08] | 0.01 [-0.06 to 0.08] | 0.01 [-0.06 to 0.08] | |


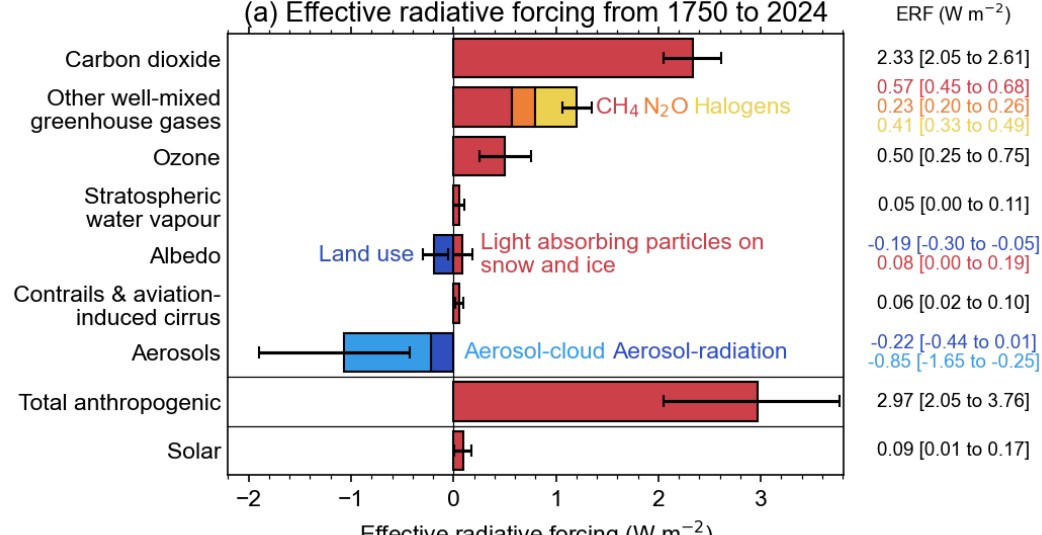


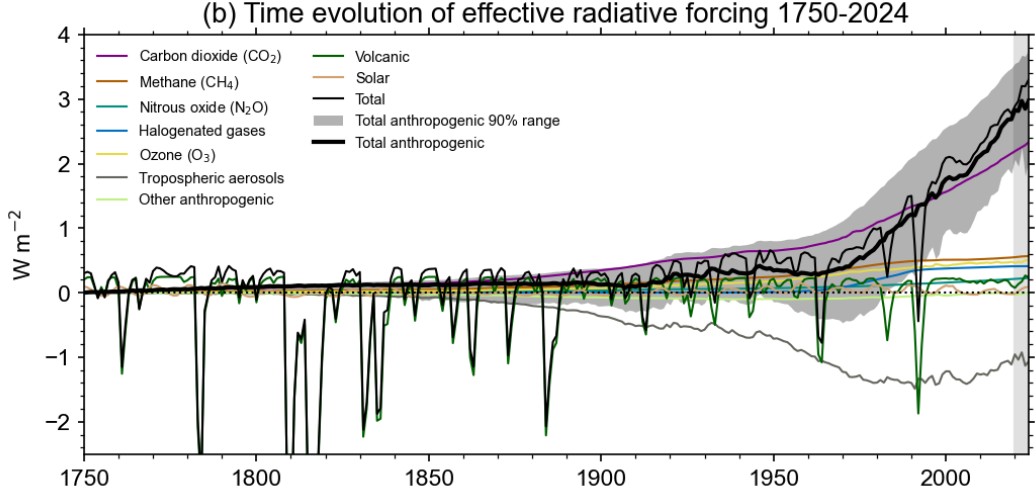



**Figure 5 Effective radiative forcing (ERF) from 1750–2024. (a) 1750–2024 change in ERF, showing best estimates (bars)**
**and 5 %–95 % uncertainty ranges (lines) from major anthropogenic components to ERF, total anthropogenic ERF and**
**solar forcing. Note that solar forcing in 2024 is a single-year estimate and hence differs from Table 3. (b) Time evolution of**
**ERF from 1750 to 2024. Best estimates from major anthropogenic categories are shown along with solar and volcanic**
**forcing (thin coloured lines), total (thin black line), and anthropogenic total (thick black line). The 5 %–95 % uncertainty**
**in the anthropogenic forcing is shown by grey shading.**

Total anthropogenic ERF has increased to 2.97 [2.05 to 3.76] $W\,m^{-2}$ in 2024 relative to 1750, compared to 2.72 [1.96
to 3.48] $W\,m^{-2}$ for 2019 relative to 1750 in AR6. The ERF has increased considerably from the 2023 estimate of 2.79
[1.79 to 3.61] $W\,m^{-2}$. 2023 was a year associated with high biomass burning aerosol which resulted in a stronger
negative aerosol forcing than recent trends. Biomass burning was also high in 2024, but lower than 2023 levels.
Sulphur emissions from shipping have declined since 2020, weakening the aerosol ERF and adding around +0.1
$W\,m^{-2}$ over 2020 to 2024 (see Sect. 7.2 and Supplement Sects. S5 and S7). The approach of including all biomass
burning aerosols is consistent with reporting ERF based on concentration increase of GHGs independent of whether
$CO_2$ and $CH_4$ are caused by anthropogenic emissions or a smaller part is caused by any feedbacks such as from biomass
burning fires or wetlands. Changes in mineral dust and sea salt are not easily relatable to human activity and are not
included in the ERF of aerosols.

The ERF from well-mixed GHGs is 3.54 [3.22 to 3.85] $W\,m^{-2}$ for 1750–2024, of which 2.33 $W\,m^{-2}$ is from $CO_2$,
0.57 $W\,m^{-2}$ from $CH_4$, 0.23 $W\,m^{-2}$ from $N_2O$ and 0.41 $W\,m^{-2}$ from halogenated gases. This is an increase of around
7% from 3.32 [3.03 to 3.61] $W\,m^{-2}$ for 1750–2019 in AR6. ERFs from $CO_2$, $CH_4$ and $N_2O$ have all increased since
the AR6 WG1 assessment for 1750–2019, owing to increases in atmospheric concentrations.






The total aerosol ERF (sum of the ERF from aerosol–radiation interactions (ERFari) and aerosol–cloud interactions
(ERFaci)) for 1750–2024 is −1.07 [−1.90 to −0.43] W m⁻² compared to −1.18 [−2.10 to −0.49] W m⁻² for 1750–2023
(Forster et al., 2024) and −1.06 [−1.71 to −0.41] W m⁻² assessed for 1750–2019 in AR6 WGI. Attributing year-to-
year trends to aerosol forcing is problematic due to the variability in biomass burning emissions. Increasing biomass
burning emissions since AR6 have been mostly offset by a decrease in emissions from energy and industrial sectors,
leading to best estimates of ERFari and ERFaci that are virtually unchanged from the 1750–2019 AR6 assessment to
the 1750–2024 determination here (Table 3).

Ozone ERF is determined to be 0.50 [0.25 to 0.75] W m⁻² for 1750–2024, slightly higher than the AR6 assessment of
0.47 [0.24 to 0.71] W m⁻² for 1750–2019. This is due to the increase in emissions of some of its precursors (CO, VOC,
$CH_4$), but this result is highly uncertain since consolidated ozone trends are not yet released. Stratospheric water
vapour from methane oxidation is unchanged (to two decimal places) since AR6. ERF from light-absorbing particles
on snow and ice being 0.08 [0.00 to 0.19] W m⁻² for 1750–2024, like AR6. We determine from provisional data that
aviation activity in 2024 has returned to pre-COVID levels (IATA, 2024). Therefore, ERF from contrails and contrail-
induced cirrus is the same as in AR6, at 0.06 [0.02 to 0.10] W m⁻² in 2024. The methodology to determine land-use
ERF has been updated (Sect. S5.4) but this forcing has a similar best estimate to 2023 and AR6, with a wider
uncertainty range that accounts for the separate assessment of irrigation forcing.

The headline assessment of solar ERF has not been re-assessed, at 0.01 [−0.06 to +0.08] W m⁻² from pre-industrial to
the 2009–2019 solar cycle mean (Table 3). Separate to the assessment of solar forcing over complete solar cycles, we
provide a single-year solar ERF for 2024 of +0.09 [+0.01 to +0.17] W m⁻² (Fig. 5a). This is higher than the single-
year estimate of solar ERF for 2019 (a solar minimum) of −0.02 [−0.08 to 0.06] W m⁻².

Volcanic ERF is included in the overall time series (Fig. 5b) but following IPCC convention we do not provide a
single-year estimate for 2024 given the sporadic nature of volcanoes. Alongside the time series of stratospheric aerosol
optical depth derived from proxies and satellite products, for 2022–2024 we include the stratospheric water vapour
contribution from the Hunga Tonga-Hunga Ha'apai (HTHH) eruption derived from Microwave Limb Sounder (MLS)
data. We estimate a net positive (positive forcing from stratospheric water vapour more than outweighing negative
forcing from stratospheric aerosols) forcing through 2024, though note that other studies find the net HTHH forcing
to be negative (Gupta et al., 2025) or close to zero (Schoeberl et al., 2024).

**6 Earth energy imbalance (EEI)**
EEI, assessed in Chap. 7 of AR6 WGI (Forster et al., 2021), provides a measure of accumulated surplus energy
(heating) in the climate system, and is hence an essential indicator to monitor the current and future status of global
warming. It represents the difference between the radiative forcing acting to warm the climate, and Earth's radiative
response, which acts to oppose this warming. Under stable climate conditions, i.e., in the absence of anthropogenic
climate forcing, this difference would be balanced over interdecadal time scales. Since at least 1970 there has been a
persistent imbalance in the energy flows that has led to excess energy being absorbed by the climate system (Forster
et al., 2021). On annual and longer timescales, the global Earth heat inventory changes associated with EEI are
dominated by the changes in global ocean heat content (OHC), which accounts for about 90 % of global heating since
the 1970s (Forster et al., 2021). This planetary heating results in changes in all components of the Earth system such
as sea-level rise, ocean warming, ice loss, rises in temperature and water vapor in the atmosphere, changes in ocean
and atmospheric circulation, continental warming and permafrost thawing (e.g. Cheng et al., 2022; von Schuckmann
et al., 2023a), with adverse impacts for ecosystems and human systems (Douville et al., 2021; IPCC, 2022).

On decadal timescales, changes in global surface temperatures (Sect. 5) can become decoupled from EEI by ocean
heat rearrangement processes (e.g. Palmer and McNeall, 2014; Allison et al., 2020). Therefore, the increase in the
Earth heat inventory arguably provides a more robust indicator of the rate of global change on interannual-to-decadal
timescales (Cheng et al., 2019; Forster et al., 2021; von Schuckmann et al., 2023a). AR6 WGI found increased
confidence in the assessment of change in the Earth heat inventory compared to previous IPCC reports due to
observational advances and joint closure of the energy and global sea level budgets (Forster et al., 2021; Fox-Kemper
et al., 2021).

AR6 estimated that EEI increased from 0.50 [0.32–0.69] $\mathrm{W\,m^{-2}}$ during the period 1971–2006 to 0.79 [0.52–
1.06] $\mathrm{W\,m^{-2}}$ during the period 2006–2018 (Forster et al., 2021). The contributions to increases in the Earth heat
inventory throughout 1971–2018 remained stable: 91 % for the full-depth ocean, 5 % for the land, 3 % for the
cryosphere and about 1 % for the atmosphere (Forster et al., 2021). Two recent studies demonstrated independently
and consistently that since 1960, the rate of warming of the world ocean is increasing at a relatively consistent pace
of $0.15 \pm 0.05\ \mathrm{W\,m^{-2}}$ per decade (Minière et al., 2023; Storto and Yang, 2024), while the rate of warming for the land,
cryosphere, and atmosphere has been increasing at  rate of $0.013 \pm 0.003\ \mathrm{W\,m^{-2}}$ per decade (Minière et al., 2023).
The increase in EEI over the last several decades (Fig. 6) has also been reported by Cheng et al. (2019), von
Schuckmann et al. (2020, 2023a), Loeb et al. (2021), Hakuba et al. (2021), Kramer et al. (2021), Raghuraman et al.
(2021) and Minère et al. (2023). The observed increase in EEI over the most recent period (i.e. past 2 decades) are
helping to drive exceptionally warm conditions (Sect. 7; Minobe et al., 2025). The increase in has been linked to rising
concentrations of well-mixed GHGs and recent reductions in aerosol emissions (Sect. 5; Raghuraman et al., 2021;
Kramer et al., 2021; Hansen et al., 2023), and to an increase in absorbed solar radiation associated with decreased
reflection by clouds and sea-ice and a decrease in outgoing longwave radiation (OLR) due to increases in trace gases
and water vapor (Loeb et al., 2021; Goesling et al., 2025).



We carry out an update to the AR6 estimate of changes in the Earth heat inventory based on updated observational
time series for the period 1971–2020 (Table 4 and Fig. 6). Time series of heating associated with loss of ice and
warming of the atmosphere and continental land surface are obtained from the recent Global Climate Observing
System (GCOS) initiative (von Schuckmann et al., 2023b; Adusumilli et al., 2022; Cuesta-Valero et al., 2023;
Vanderkelen and Thiery, 2022; Nitzbon et al., 2022; Kirchengast et al., 2022). We use the original AR6 time series
ensemble OHC time series for the period 1971–2018 and then an updated five-member ensemble for the period 2019–
2024. We "splice" the two sets of time series by adding an offset as needed to ensure that the 2018 values are identical.
The AR6 heating rates and uncertainties for the ocean below 2000 m are assumed to be constant throughout the period.
The time evolution of the Earth heat inventory is determined as a simple summation of time series of atmospheric
heating; continental land heating; heating of the cryosphere; and heating of the ocean over three depth layers: 0–700,
700–2000 and below 2000 m (Fig. 6a). While von Schuckmann et al. (2023a) have also quantified heating of
permafrost and inland lakes and reservoirs, these additional terms are small and not included here for consistency with
AR6 (Forster et al., 2021).

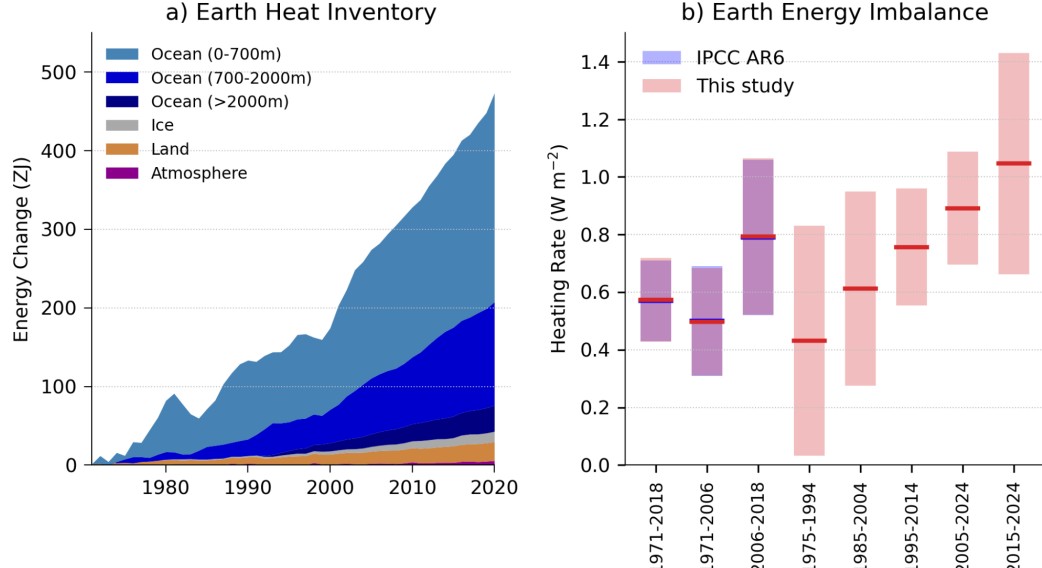





**Figure 6 (a) Observed changes in the Earth heat inventory for the period 1971–2020, with component contributions as indicated in the figure legend. (b) Estimates of the Earth energy imbalance for the IPCC AR6 assessment periods, for consecutive 20-year periods and the most recent decade. Shaded regions indicate the *very likely* range (90 % to 100 % probability). Data use and approach are based on the AR6 methods and further described in Supplement Sect. S6. For the IPCC AR6 periods our assessment closely matches that in AR6. Note the periods in our assessment overlap with different IPCC AR6 periods.**

In our updated analysis, we find successive increases in EEI for each 20-year period since 1975, with an estimated value of 0.43 [0.03 to 0.83] $\text{W m}^{-2}$ during 1975–1994 that more than doubled to 0.89 [0.7 to 1.09] $\text{W m}^{-2}$ during 2005–2024 (Fig. 6b). In addition, there is some evidence that the warming signal is propagating into the deeper ocean over time, as seen by a robust increase of ocean warming in the 700-2000m depth layer since the 1990s (von Schuckmann et al., 2020; 2023; Cheng et al., 2019, 2022). The model simulations qualitatively agree with the observational evidence (e.g. Gleckler et al., 2016; Cheng et al., 2019), further suggesting that more than half of the OHC increase since the late 1800s occurs after the 1990s.

The update of the AR6 assessment periods to end in 2024 results in systematic increases of EEI: 0.68 $\text{W m}^{-2}$ during 1977–2024 compared to 0.57 $\text{W m}^{-2}$ during 1971–2018; and 0.99 $\text{W m}^{-2}$ during 2012–2024 compared to 0.79 $\text{W m}^{-2}$ 2006–2018 (Table 4). The trend and interannual variability of EEI can largely be explained by a combination of surface temperature changes and radiative forcing (Hodnebrog et al., 2024). However, there was a jump in 2023 and 2024 which is still being investigated (see Sect. 7.2), but which is also discussed in the light of recent exceptional extreme climate conditions (Minobe et al., 2025).

**Table 4 Estimates of the Earth energy imbalance (EEI) for AR6 and the present study.**

| Time Period | Earth energy imbalance (W m⁻²). Square brackets [show 90% confidence intervals]. | |
| --- | --- | --- |
| | **IPCC AR6** | **This Study** |
| 1971-2018 | 0.57 [0.43 to 0.72] | 0.57 [0.43 to 0.72] |
| 1971-2006 | 0.50 [0.32 to 0.69] | 0.50 [0.31 to 0.68] |
| 2006-2018 | 0.79 [0.52 to 1.06] | 0.79 [0.52 to 1.07] |
| 1977-2024 | - | 0.68 [0.52 to 0.85] |





| 2012-2024 | - | 0.99 [0.70 to 1.28] |
|---|---|---|

590

## 7 Observed surface temperature change

### 7.1 Change since 1850-1900

AR6 WGI Chap. 2 assessed the 2001–2020 globally averaged surface temperature change above an 1850–1900 baseline to be 0.99 [0.84 to 1.10] °C and 1.09 [0.95 to 1.20] °C for 2011–2020 (Gulev et al., 2021). Updated estimates to 2013-2022 of 1.15 [1.00–1.25] °C were given in AR6 SYR (Lee et al., 2023), matching the estimate in Forster et al. (2023).

There are choices around the methods used to aggregate surface temperatures into a global average, how to correct for systematic errors in measurements, methods of infilling missing data, and whether surface measurements or atmospheric temperatures just above the surface are used. These choices, and others, affect temperature change estimates and contribute to their uncertainty (IPCC AR6 WGI Chap. 2, Cross Chap. Box 2.3, Gulev et al., 2021). The methods chosen here closely follow AR6 WGI and are presented in the Supplement Sect. S7. Confidence intervals are taken from AR6 as only one of the employed datasets regularly updates ensembles (see Supplement Sect. S7).

Based on the updates available as of March 2025, the change in global surface temperature from 1850–1900 to 2015–2024 is presented in Fig. 7. These data, using the same underlying datasets (with some version changes: see Supplement Sect. S7) and methodology as AR6, estimate 1.24 [1.11–1.35] °C of warming, an increase of 0.15 °C within four years from the 2011–2020 value reported in AR6 WGI (Table 5), or 0.14 °C from the 2011–2020 value in the most recent dataset version. The decade 2015-2024 was 0.31 °C warmer than the previous decade (2005–2014). These changes, although amplified somewhat by the exceptionally warm years in 2023 and 2024, are broadly consistent with typical warming rates over the last few decades, which were assessed in AR6 as 0.76 °C over the 1980–2020 period (using ordinary-least-square linear trends) or 0.019 °C per year (Gulev et al., 2021). They are also broadly consistent with projected warming rates from 2001–2020 to 2021–2040 reported in AR6, which have a very likely range between 0.016 °C per year and 0.036 °C per year under SSP2-4.5 (Lee et al., 2021, their Table 4.5), and with human-induced warming rates discussed in Sect. 8.4.

Land temperatures have increased by 1.79 [1.56–2.03] °C from 1850-1900 to 2015-2024, and ocean temperatures by 1.02 [0.81-1.13] °C over the same period, implying that most land areas have already experienced more than 1.5 °C of warming from the 1850–1900 period. As was the case for the periods reported in AR6, the ratio of observed land



to ocean warming is in the vicinity of 1.75, somewhat higher than the ratio of 1.5 [1.4–1.7] projected by the end of the
century in CMIP6 models (AR6, their Table 4.2 and Section 4.5.1.1.1). The additional observed warming since 2020
in the most recent dataset versions (0.21 °C for land, 0.13 °C for ocean) has a ratio within the CMIP6 projections
range.

**Table 5 Estimates of global surface temperature change from 1850–1900 [*very likely* (90 %–100 % probability) ranges] for**
**IPCC AR6 and the present study.**

| Time period | Temperature change from 1850-1900 (°C) | |
|---|---|---|
|  | IPCC AR6 (as reported) | This study |
| Global, most recent 10 years | 1.09 [0.95 to 1.20] (to 2011-2020) | 1.24 [1.11 to 1.35] (to 2015-2024) |
| Global, most recent 20 years | 0.99 [0.84 to 1.10] (to 2001-2020) | 1.09 [0.93 to 1.20] (to 2005-2024) |
| Land, most recent 10 years | 1.59 [1.34 to 1.83] (to 2011-2020) | 1.79 [1.56 to 2.03] (to 2015-2024) |
| Ocean, most recent 10 years | 0.88 [0.68 to 1.01] (to 2011-2020) | 1.02 [0.81 to 1.13] (to 2015-2024) |


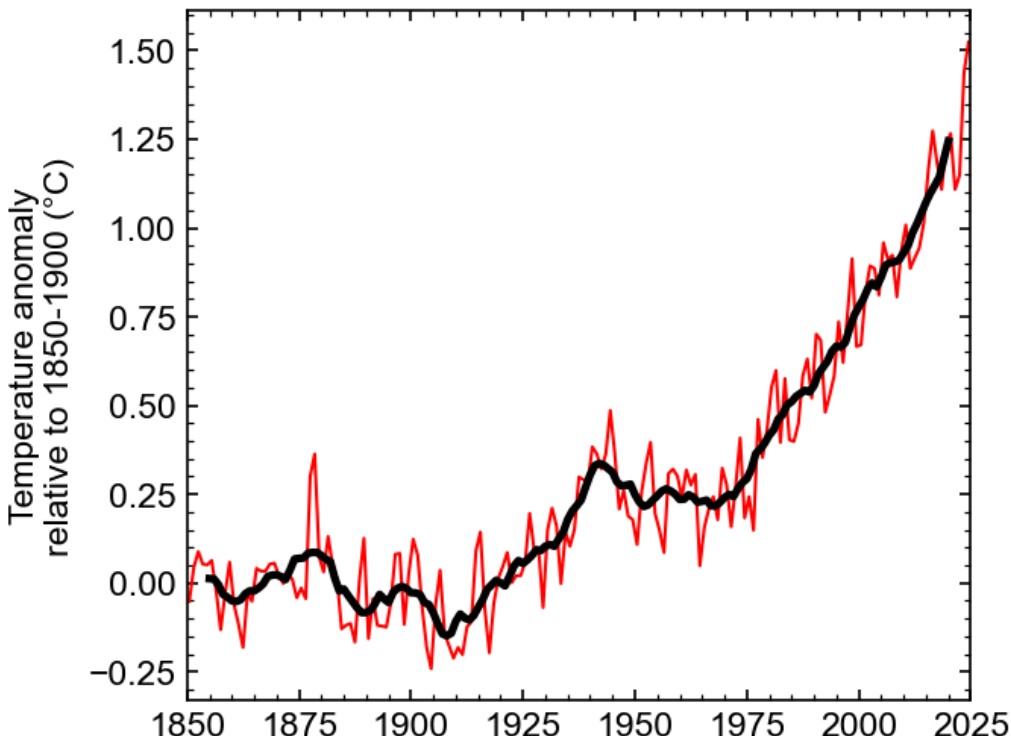


**Figure 7 Annual (thin line) and decadal (thick line) means of global surface temperature (expressed as a change from the 1850–1900 reference period). Temperatures are based on an average of four datasets following AR6, see Supplement Sect. S7 for details.**

**7.2 2023-2024 global mean temperature -anomalies**

At the time, 2023 set a new global annual-mean surface temperature change record, with a best estimate of 1.44 °C, beating 2016 by 0.16 °C. 2024 surpassed this, reaching a best estimate of 1.52 °C; 2024, becoming the first calendar year since preindustrial likely exceeding 1.5 °C (Fig. 7). Natural drivers and internal variability are expected to modulate human-caused warming at interannual-to-decadal timescales. 2024 is assessed to be 0.16 °C warmer than the updated human-induced value (Table 6) while 2022 was 0.06 °C colder. These values are not inconsistent with AR6, which estimated the effect of internal variability in any single year be +/- 0.25 °C based on CMIP6 models, nor with the lower estimated ranges (+/- 0.17 °C) when calculated from observational products (Trewin, 2022).

The probability of seeing an observed temperature of 1.52 °C in 2024 considering a human-induced warming equal to 1.36 °C is about 1 chance out of 6 (Fig. 8a). The methodology to calculate this probability consists in comparing the GSAT observed anomaly to those expected from CMIP6 models following the framework adopted in AR6 in Chapter





3 (Eyring et al., 2021) for decadal trends and adapted here for interannual time scale issues. The same probability but
conditional to the fact that 2024 followed an El Niño year and that the Atlantic Multidecadal Variability (AMV) was
in a positive phase (Supplement Sect. S7), rises to 1 chance out of 2. 2024 can therefore be treated as a "normal" year,
i.e. very much expected at the actual human-caused global warming level when the internal modes of variability are
taken into account and when assessed from a very large number of simulations from large ensembles. Based on the
same calculation, we estimate that a year as warm as 2023 would occur once in 4 years at human-induced warming
equal to 1.31 °C (Fig. 8b). It drops to 1-in-14 [10-20, CI 5-95%] year event, i.e. a rare-to-exceptional event, when
considering that 2023 followed a La Nina year and despite persistent positive AMV. Within such a framework, 2022,
that was colder than human-induced warming, could be interpreted as a normal/expected year considering that 2021
was a La Nina year and AMV positive (Fig. 8c).


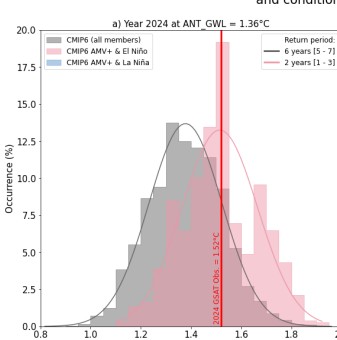
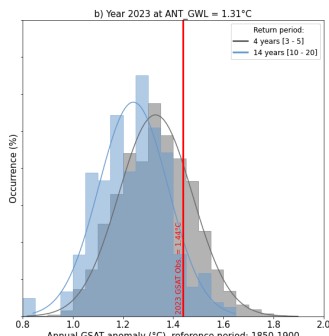
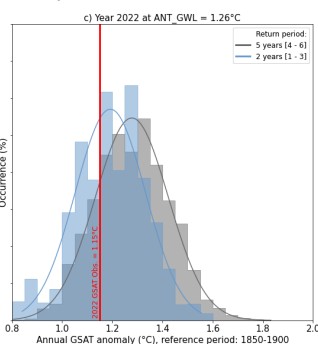







**Figure 8 a) Gray histograms of global surface air temperature (GSAT) interannual anomalies estimated from 15 CMIP6 models extracted from all available SSP scenarios (~700 members) at anthropogenic global warming levels (ANT_GWL) corresponding to a) 2024, b) 2023, c) 2022. The red vertical bar stands for the observational consolidated GSAT annual anomalies (Sect. 7.1). The return period of the observed annual GSAT event estimated from the CMIP6 distribution is provided (upper-corner). Associated [5-95%] likely range is assessed through bootstrapping. Interannual anomalies are obtained following Trewin (2022) method over 10-yr sliding windows. Only models providing large-ensembles (n members >5) and having at least one member whose interannual variance of GSAT is compatible with observational estimates, are selected. Colored histograms stand for the same distribution but conditional to the combined phase of El Nino Southern Oscillation (ENSO) and Atlantic Multidecadal Variability (AMV). SST Anomalies for the modes of variability are calculated from the residual of SST obtained after removing the modelled forced response estimated as model ensemble mean. A year is considered as an El Nino/La Nina year if the (October-December) Oceanic Nino Index (ONI) index of the previous year is greater/lower than one standard deviation. A year is considered as an AMV+ year if the annual North Atlantic average SST is greater than one standard deviation. Light pink represents years when ONI and AMV are concomitantly positive and light-blue when ONI is negative.**

The increase in global temperature between 2022 and 2023 and in particular in global sea surface temperature is exceptional based on model estimates accounting for projected known human and natural forcings plus internal variability (Rantanen and Laaksonen, 2024; Terhaar et al., 2025, Cattiaux et al., 2024). The La Niña-to-El Niño sequence is of key importance and has been likely reinforced by enhanced energy uptake due to multi-year persistence in the preceding La Niña. The temporal synchronicity between the modes of variability in all basins is hypothesized to have played a role in the jump (Minobe et al., 2025) with the North Atlantic being record warm (Guinaldo et al., 2025) and the austral sea ice extent being record low (Purish and Doddridge, 2023).

Possible specific causes beyond internal variability, many of which are already accounted for in the estimated human-induced warming level, have been postulated e.g.: International Maritime Organization rules on shipping fuel sulphur content that came into force in January 2021; the eruption of Hunga Tonga Hunga Ha'apai in January 2022 and other subsequent smaller volcanic activity; and a faster-than-expected onset of Solar Cycle 25 (see Supplement Section S7 for details and references). A key diagnostic of these changes including both external forcing and internal variability was the exceptional magnitude of the net energy increase into the Earth system from mid-2022 to mid-2023, driven in large part by the reduced reflectance and greater absorption of solar radiation (Hodnebrog et al., 2024; Goessling et al., 2024; Minobe et al. 2025), which may be influenced by cloud feedbacks (Tselioudis et al., 2024) as well as surface reflectance and atmospheric composition change (see also Sect. 6).

Our analysis, detailed in Supplement Sect. S7, makes use of estimates of variability and radiative forcing contributions and their uncertainty based on Sect 5. and the published literature. It shows that the increase in 2023 and 2024 compared to previous years could be explained by a combination of factors. In summary, our analyses show that, although the relative weight between the physical processes in explaining the high surface temperatures remain to be



better quantified, the 2023 and 2024 observed temperatures are not inconsistent with the level of human induced
warming assessed next, in Sect. 8.

**8 Human contribution to surface temperature change**

Human-induced warming, also known as anthropogenic warming, refers to the component of observed global surface
temperature increase attributable to both the direct and indirect effects of human activities, which are typically grouped
as follows: well-mixed GHGs (consisting of $CO_2$, $CH_4$, $N_2O$ and F-gases) and other human forcings (consisting of
aerosol–radiation interaction, aerosol–cloud interaction, black carbon on snow, contrails, ozone, stratospheric $H_2O$
and land use) (Eyring et al., 2021). The remaining contributors to total warming are natural: consisting of both natural
forcings (such as solar and volcanic activity) and internal variability of the climate system (such as variability related
to El Niño/La Nina events).

An assessment of human-induced warming was provided in two reports within the IPCC's Sixth Assessment cycle:
first in SR1.5 in 2018 [Chap. 1 Sect. 1.2.1.3 and Fig. 1.2 (Allen et al., 2018), summarised in the Summary for
Policymakers (SPM) Sect. A.1 and Fig. SPM.1 (IPCC, 2018)] and second in AR6 in 2021 [WGI Chap. 3 Sect. 3.3.1.1.2
and Fig. 3.8 (Eyring et al., 2021), summarised in the WGI Summary for Policymakers (SPM) Sect. A.1.3 and Fig.
SPM.2 (IPCC, 2021b)], and quoted again without any updates in SYR [Sect. 2.1.1 and Fig. 2.1 (IPCC,2023a) and
SYR Summary for Policymakers (SPM) Sect. A.1.2. (IPCC 2023b)].

**8.1 Warming period definitions in the IPCC Sixth Assessment cycle**

Temperature increases are defined relative to a baseline; IPCC assessments typically use the 1850–1900 average
temperature as a proxy for the climate in pre-industrial times, referred to as the period before 1750, even though a
small amount of warming likely occurred over 1750-1850 (see AR6 WGI Cross Chapter Box 1.2). Temperatures in
the IPCC were reported as either GMST or GSAT, see Supplement Sect. 8.1 for details.

Tracking progress towards the long-term global goal to limit warming, in line with the Paris Agreement, requires the
assessment of both what the current level of global surface temperatures are and whether a level of global warming,
such as 1.5 °C, is being reached. Definitions for these were not specified in the Paris Agreement, and several ways of
tracking levels of global warming are in use; here we focus on those adopted within AR6. When determining whether
warming thresholds have been passed, both AR6 and SR1.5 adopted definitions that depend on future warming; in
practice, levels of current warming were therefore reported in AR6 and SR1.5 using additional definitions that
circumvented the need to wait for observations of the future climate, as described next. AR6 defined crossing-time for
a level of global warming as the midpoint of the first 20-year period during which the average *observed* warming for
that period exceeds that level of warming (see AR6 WGI Chapter 2 Box 2.3) (the level of warming for a given year
defined in this way is therefore not known until 10 years after that year). AR6 therefore reported current levels of both



*observed* and *human-induced* warming as their averages over just the most recent 10 years (which gives warming that
lags by only 5 years instead of 10 years) (see AR6 WGI Chapter 3 their Sect. 3.3.1.1.2); we refer to this definition as
the "AR6 decade-average" warming. SR1.5 defined the level of warming in a given year as the average *human-induced*
warming, in GMST, of a 30-year period centred on that year; when the given year is the *current* year, SR1.5 specified
that the future 15 years (required for the mean) are revealed by extrapolating the multidecadal trend (see SR1.5 Chapter
1, their Sect. 1.2.1); we refer to this definition as the "SR1.5 trend-based" warming. If the multidecadal trend is
interpreted as being linear (which it has been very close to over recent decades), this definition of current warming is
equivalent to the end-point of the trend line through the most recent 15 years of human-induced warming, and therefore
provides a definition of warming for the current year that depends only on historical warming. This interpretation
produces results that in recent years have been identical (or extremely close) to the current annual mean value of
human-induced warming (see results in Sect. 8.2, and Supplement Sect. S8.3), so in practice the attribution assessment
in SR1.5 was based not on the trend-based definition, but on the simple annual-year attributed warming; we refer to
this definition as the "SR1.5 annual-mean" warming. A diagram of these three definitions is given in Supplement Fig.
S11.

**8.2 Updated assessment approach of human-induced warming to date**
This paper provides an update of the AR6 WGI and SR1.5 human-induced warming assessments including, for
completeness, all three definitions (AR6 decade-average, SR1.5 trend-based, and SR1.5 annual-mean). The 2024
updates in this paper follow the same methods and process as the 2022 and 2023 updates provided in Forster et al.
(2023, 2024). Global mean surface temperature (GMST) is adopted as the definition of global surface temperature
(see Supplement Sect. S8.1). The three attribution methods used in AR6 are retained: the Global Warming Index
(GWI) (building on Haustein et al., 2017), regularised optimal fingerprinting (ROF) (as in Gillett et al., 2021) and
kriging for climate change (KCC) (Ribes et al., 2021). Details of each method, their different uses in SR1.5 and AR6,
and any methodological changes, are provided in Supplement Sect. S8.2; method-specific results are also provided in
Supplement Sect. S8.3. The overall estimate of attributed global warming for each definition (decade-average, trend-
based, and annual-mean), is based on a multi-method assessment of the three attribution methods (GWI, KCC, ROF);
the best estimate is given as the 0.01 °C-precision mean of the 50th percentiles from each method, and the *likely* range
is given as the smallest 0.1 °C-precision range that envelops the 5th to 95th percentile ranges of each method. This
assessment approach is identical to last year's update (Forster et al. (2024)); it is directly traceable to and fully
consistent with the assessment approach in AR6, though it has been lightly extended in ways that are explained in
Supplement Sect. S8.4.




Results are summarised in Table 6 and Fig. 9. Method-specific contributions to the assessment results, along with time
series, are given in the Supplement, Sect. S8.3. Where results reported in GSAT differ from those reported in GMST
(see Supplement Sect. S8.1), the additional GSAT results are given in Supplement Sect. S8.3.

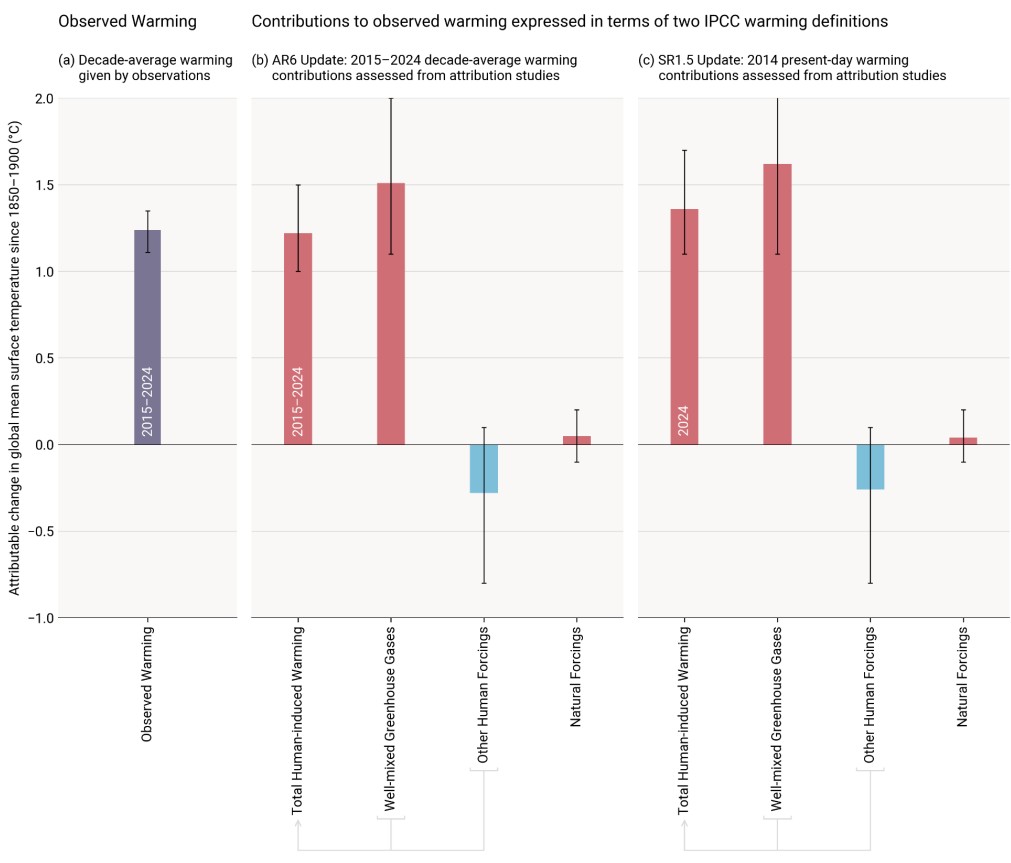




**Figure 9 Updated assessed contributions to observed warming relative to 1850–1900; see AR6 WGI SPM.2. Results for all**
**time periods in this figure are calculated using updated datasets and methods. The 2015–2024 average and 2024 results are**
**this year's updated assessments for AR6 and SR1.5, respectively. Panel (a) shows updated observed global warming from**
**Sect. 7, expressed as total global mean surface temperature (GMST), due to both anthropogenic and natural influences.**
**Whiskers give the "*very likely*" range. Panels (b) and (c) show updated assessed contributions to warming, expressed as**
**global mean surface temperature (GMST), from natural forcings and total human-induced forcings, which in turn consist**
**of contributions from well-mixed GHGs and other human forcings. Whiskers give the "*likely*" range. Changes to warming**
**levels since the IPCC sixth assessment cycle are depicted in Supplement Fig. S10.**
**Table 6 Updates to assessments in the IPCC 6th assessment cycle of warming attributable to multiple influences. Estimates**
**of warming attributable to multiple influences, in °C, relative to the 1850–1900 baseline period. Results are given as best**
**estimates, with the *likely* range in brackets, and reported as global mean surface temperature (GMST).  Results from the**
**IPCC 6th assessment cycle, for both AR6 and SR1.5, are quoted in columns labelled (i) and are compared with repeat**
**calculations in columns labelled (ii) for the same period using the updated methods and datasets to see how methodological**
**and dataset updates alone would change previous assessments. Assessments for the updated periods are reported in columns**
**labelled (iii). * Updated GMST observations, quoted from Sect. 7 of this update, are marked with an asterisk, with "*very***
***likely*" ranges given in brackets. ** In AR6 WGI, best-estimate values were not provided for warming attributable to well-**
**mixed GHGs, other human forcings and natural forcings (though they did receive a "*likely*" range); for comparison, best**
**estimates (marked with two asterisks) have been retrospectively calculated in an identical way to the best estimate that AR6**
**provided for anthropogenic warming (see discussion in Supplement Sect. S8.4.1). *** The SR1.5 assessment drew only on**
**GWI rounded to 0.1°C precision, whereas the repeat and updated calculations use the updated multi-method assessment**
**approach.**

| Estimates of warming attributable to multiple influences, in °C, relative to the 1850–1900 baseline period | | | | | | |
|---|---|---|---|---|---|---|
| Results are given as best estimates, with the *likely* range in brackets, and reported as Global Mean Surface Temperature (GMST). | | | | | | |
| Definition ➡ | **(a) IPCC AR6 Attributable Warming Update** | | | **(b) IPCC SR1.5 Attributable Warming Update** | | |
| | *Value for decade (average of previous 10-year period)* | | | *Value for single year (30-year mean centred on current year)* | | |
| Period ➡ <br><br> Component ⬇ | **(i) 2010-2019** *Quoted from AR6 Chapter 3 Sect. 3.3.1.1.2 Table 3.1* | **(ii) 2010-2019** *Repeat calculation using the updated methods and datasets* | **(iii) 2015-2024** *Updated value using updated methods and datasets* | **(i) 2017** *Quoted from SR1.5 Chapter 1 Sect. 1.2.1.3* | **(ii) 2017** *Repeat calculation using the updated methods and datasets* | **(iii) 2024** *Updated value using updated methods and datasets* |
| **Observed** | 1.06 [0.92 to 1.17] | 1.07  [0.89 to 1.22] * | 1.24 [1.11 to 1.35] * | - | - | 1.52 |
| **Anthropogenic** | 1.07 [0.8 to 1.3] | 1.09 [0.9 to 1.3] | 1.22 [1.0 to 1.5] | 1.0 [0.8 to 1.2] *** | 1.13 [0.9 to 1.3] | 1.36 [1.1 to 1.7] |
| **Well-mixed GHGs** | 1.40** [1.0 to 2.0] | 1.40 [1.0 to 1.9] | 1.51 [1.1 to 2.0] | N/A | 1.45 [1.0 to 1.9] | 1.62 [1.1 to 2.1] |
| **Other human forcings** | -0.32** [-0.8 to 0.0] | -0.30 [-0.8 to 0.0] | -0.28 [-0.8 to 0.1] | N/A | -0.31  [-0.8  to 0.1] | -0.26 [-0.8 to 0.1] |
| **Natural forcings** | 0.03** [-0.1 to 0.1] | 0.05 [-0.1 to 0.2] | 0.05 [-0.1 to 0.2] | N/A | 0.05 [-0.1 to 0.2] | 0.04 [-0.1 to 0.2] |




The repeat calculations for attributable warming in 2010–2019 exhibit good correspondence with the results in AR6
WGI for the same period (see also Supplement, Sect. S8). The repeat calculation for the level of attributable
anthropogenic warming in 2017 is about 0.1 °C larger than the estimate provided in SR1.5 for the same period,
resulting from changes in methods and observational data (see AR6 WGI Chapter 2 Box 2.3). The updated results for
warming contributions in 2024 are higher than in 2017 due also to 7 additional years of increasing anthropogenic
forcing. Note also that the SR1.5 assessment only used the GWI method, whereas these annual updates apply the full
AR6 multi-method assessment (see Supplement Sect. S8.4 for details and rationale).

In this 2025 update, we assess the 2015–2024 decade average human induced-warming at 1.22 [1.0 to 1.5] °C, which
is 0.15°C above the AR6 assessment for 2010–2019. The single year average human-induced warming is assessed to
be 1.36 [1.1 to 1.7] °C in 2024 relative to 1850–1900. In general, these forced warming levels have evolved steadily
and predictably in line with the current warming rate within uncertainty. The uncertainty range for the single-year
level of anthropogenic warming already included 1.5 °C in previous years' assessments, and for the first time this year
also lies at the edge of the uncertainty range for the (lagged) decade mean definition. The single-year  anthropogenic
warming best estimate is well below the observed best estimate for 2024 (1.52 °C, see Sect. 7), but note that the best
estimate and lower uncertainty for observed warming lies within the uncertainty for single-year anthropogenic
warming from each of the three attribution methods (see Supplement Table S5), whereas the upper uncertainty range
of observed warming lies above the range for anthropogenic warming for the two attribution methods that fully exclude
internal variability.

The best estimates for decade-average and single-year human-induced warming are 0.04 °C and 0.05 °C respectively
above the value estimated in the previous update for the year 2023 (Forster et al., 2024), but should not be interpreted
as a substantive increase in the rate of forced anthropogenic warming,  as the rate increase is well within uncertainty
ranges (Sect. 8.3).

AR6 found that, averaged for the 2010–2019 period, essentially all observed global surface temperature change was
human-induced, with solar and volcanic drivers and internal climate variability making a negligible contribution. This
conclusion remains the same for the 2015–2024 period. Generally, whatever methodology is used, on a global scale,
the best estimate of the current level of human-induced warming is (within small uncertainties) similar to the observed
global surface temperature change (Table 6).

**8.3 Rate of human-induced global warming**
Estimates of the human-induced warming rate follow the same methodology as in the previous year's update (a rolling
10-year linear trend in attributed anthropogenic warming). A full description of the approach can be found in the
Supplement Sect. S8.5. The rate of increase in attributed anthropogenic warming over time is distinct from the rate of
increase in the observed global surface temperature, which is also affected by internal variability such as El Niño and
natural forcings such as volcanic activity (see discussions in Sect. 7.2). The rate of anthropogenic warming we estimate
here is driven by the rate of change of anthropogenic ERF (Sect. 5), with variations in the climate forcing trend over
time correlating with variations in the rate of attributed warming (Fig. 10).

**Decadal rates of change for contributions to Attributed Global Warming and Effective Radiative Forcing**



**Figure 10 Rates of (a) attributable warming (global mean surface temperature (GMST)) and (b) effective radiative forcing.**
**The attributable warming rate time-series are calculated using the Global Warming Index method with full ensemble**
**uncertainty. The observed GMST rates included for reference are also calculated with uncertainty from the HadCRUT5**
**ensemble, and, for consistency with the attributed warming rates, do not include standard regression error, which, for**
**observed warming, would increase the size of the error bars. The effective radiative forcing rates are calculated using a**
**representative 1000-member ensemble of the forcings provided in Sect. 5 of this paper. The depicted rates are the decadal**
**rates, with the end year of the decade in question being the value given on the time axis.**

Estimates for the trend derived from the three warming attribution methodologies are presented in Table 7, with results
for individual attribution methods detailed in the Supplement Table S6. The GWI (based on observed warming and
forcing) and KCC (based on CMIP simulations) methodologies report results that are in close agreement, while
estimates derived with the ROF method (also based on CMIP simulations) are more strongly influenced by residual
internal variability that remains in the anthropogenic warming signal due to the limitations in size of the CMIP
ensemble. The median result is presented at 0.01 °C/decade precision for the overall multi-method rate of warming
assessment.

An overall best estimate attributed rate of human-induced warming of 0.27 °C/decade is found for the decade 2015–
2024. This increased rate relative to the 0.2 °C/decade AR6 assessment is broken down in the following way: (i) 0.03



°C/decade from changing the rounding precision (updating the AR6 2010-2019 warming rate assessment from 0.2 to
0.23 °C/decade), (ii) 0.03 °C/decade is due to methodological and dataset updates (updating the 2010–2019 warming
rate from 0.23 °C/decade to 0.26 °C/decade; including the effect of adding 5 additional observed years to the attribution
over the entire historical period), and (iii) 0.01 °C/decade due to a real increase in rate for the 2015–2024 period since
the 2010–2019 period (updating 0.26 °C/decade for 2010–2019 to 0.27 °C/decade for 2015–2024), consistent with
increased GHG emissions over the last decade. The spread of rates across the three attribution methods remains similar
to their spread in AR6, and previous updates of this work, and hence does not support a decrease in the headline
uncertainty range. However, as previous assessments suggested, we update the uncertainty range for the rate of human-
induced warming from [0.1–0.3] °C/decade in AR6 to [0.2–0.4] °C/decade to better reflect the closer agreement of
the 5% floors and the larger spread in the 95% ceilings of the three methods, and higher rate from the ROF method.
The rate of human-induced warming for the 2015-2024 decade is concluded to be 0.27 °C/decade with a range of
[0.2–0.4] °C/decade). This agrees with the decadal trend in observed warming of 0.26 °C per decade (also calculated
as a linear trend through 10-year periods - see Sect. 7.1). It is important to note, however, that internal variability leads
to the decadal rates of observed warming being far less stable than for anthropogenic warming, and the very close
correspondence between the two this year is somewhat incidental (see Fig. 10).

**Table 7 Updates to the IPCC AR6 rate of human-induced warming. Results for each method are given in the Supplement**
**Table S6; assessment results are given as a best estimate with *likely* range in brackets. Results from AR6 WGI (Ch.3 Sect.**
**3.3.1.1.2 Table 3.1) are quoted in column (i), and compared with a repeat calculation using the updated methods and**
**datasets in column (ii), and finally updated for the 2015-2024 period in column (iii). The AR6 assessment result was identical**
**to the SR1.5 assessment result, though the latter was based on a different set of studies and timeframes. * Note that for**
**clarity and ease of comparison with this year's updated assessment, in the assessed rate in column (i) both quotes the**
**assessment from AR6 and retrospectively applies the median approach adopted in this paper. The observed rates are**
**calculated using the multi-dataset observed temperature dataset from Sect. 7; no ensemble is available for this, hence the**
**absence of an uncertainty range.**

| Estimates of anthropogenic warming rate, in °C per decade Results are given as best estimates, with brackets giving the *likely* range for the assessments, and 5-95% uncertainty for the individual methods | | | |
|---|---|---|---|
| Definition ➡ | **IPCC AR6 Anthropogenic Warming Rate Update** *Linear trend in anthropogenic warming over the trailing 10-year period* | | |
| Period ➡ | **(i) 2010-2019** *Quoted from AR6 Chapter 3 Sect. 3.3.1.1.2 Table 3.1* | **(ii) 2010-2019** *Repeat calculation using the updated methods and datasets* | **(iii) 2015-2024** *Updated value using updated methods and datasets* |
| **Anthropogenic Warming Rate Assessment** | Quoted from AR6: 0.2 [0.1 to 0.3]  Using the median approach: 0.23 [0.1 to 0.3] * | 0.26 [0.2 to 0.4] | 0.27 [0.2 to 0.4] |
| **Observed** | | 0.37 | 0.26 |







**9 Remaining Carbon Budget**
AR5 (IPCC, 2013) assessed that long-term global surface temperature increase caused by $CO_2$ emissions is close to
linearly proportional to the total amount of cumulative $CO_2$ emissions (Collins et al., 2013). The most recent AR6
report reaffirmed this assessment and highlights that this near-linear relationship also holds between cumulative $CO_2$
emissions and maximum global surface temperature increase caused by $CO_2$ (Canadell et al., 2021). This near-linear
relationship implies that for keeping global warming below a specified temperature level, one can estimate the total
amount of $CO_2$ that can ever be emitted. When expressed relative to a recent reference period, this is referred to as the
remaining carbon budget (Rogelj et al., 2018).

AR6 assessed the remaining carbon budget (RCB) in Chap. 5 of its WGI report (Canadell et al., 2021) for warming
limits ranging from 1.3 to 2.4 °C relative to the 1850-1900 period (see Table 5.8 in Canadell et al., 2021). A selection
of these (1.5, 1.7, and 2 °C) were also reported in its Summary for Policymakers (Table SPM.2, IPCC, 2021b). These
RCB values are updated in this section using the same method as last year (Forster et al., 2024). Data for four warming
limits (1.5, 1.6, 1.7 and 2 °C) are included in Table 8 while figures for more values are included in the Supplement
Sect. S9.

The RCB is estimated by application of the WGI AR6 method described in Rogelj et al. (2019), which involves the
combination of the assessment of five factors: (i) the amount of human-induced warming for the most recent decade
(given in Sect. 8), (ii) the transient climate response to cumulative emissions of $CO_2$ (TCRE), which quantifies the
linear proportionality between cumulative $CO_2$ emissions and $CO_2$-induced warming (iii) the zero emissions
commitment (ZEC), representing the expected amount of additional (at present unrealized) warming caused by past
$CO_2$ emissions (iv) the temperature contribution of future non-$CO_2$ emissions and (v) an adjustment term for Earth
system feedbacks that are otherwise not captured through the other factors. AR6 WGI reassessed all five terms
(Canadell et al., 2021). Lamboll et al. (2023) further considered the temperature contribution of non-$CO_2$ emissions
and integrated different uncertainties, while Rogelj and Lamboll (2024) clarified the reductions in non-$CO_2$ emissions
that are assumed in the RCB estimation.

The RCB for 1.5, 1.6, 1.7 and 2 °C warming levels is re-assessed based on the most recent available data. Estimated
RCBs are reported in Table 8. They are expressed relative to the start of 2025 for estimates based on the 2015–2024
human-induced warming update (Sect. 8). Based on the variation in non-$CO_2$ emissions across the scenarios in AR6
WGIII scenario database, the estimated RCB values can be higher or lower by around 200 $GtCO_2$ depending on how
successful non-$CO_2$ emissions reductions are (Lamboll et al., 2023; Rogelj and Lamboll, 2024). Notably, RCB
estimates consider the subset of non-$CO_2$ emission scenarios in the AR6 WGIII database that are aligned with a global
transition to net zero $CO_2$ emissions (Lamboll et al., 2023; Rogelj and Lamboll, 2024). These estimates assume median
reductions in non-$CO_2$ emissions between 2020–2050 of $CH_4$ (about 50 %), $N_2O$ (about 20 %) and $SO_2$ (about 80 %)
(see Supplement, Sect. S9 and Table S7 and (Rogelj and Lamboll, 2024)). If these non-$CO_2$ GHG emission reductions



are not achieved, the RCB for all temperature targets would be smaller than the values reported here in Table 8 (see
Lamboll et al., 2023, Rogelj and Lamboll, 2024).

Compared to RCB values reported in AR6, our estimates here are smaller owing to several factors. First, AR6 budgets
were expressed from 2020 onwards, and approximately 200 GtCO$_2$ have been emitted between 2020 and 2024.
Second, we use updated physical models of non-CO$_2$ forcing which lead to an increased estimate of the importance of
aerosols that are expected to decline with time in low emissions pathways (Rogelj et al., 2014; Rogelj and Lamboll,
2024). This decreased negative forcing from aerosols is expected to cause additional net non-CO$_2$ warming because
more non-CO$_2$ GHG warming is being unmasked and this decreases the RCB (Lamboll et al., 2023) by slightly over
100 GtCO$_2$. There was also a small reduction in the budget (about 10 GtCO$_2$) from using the newer AR6 scenario set.
Finally, the updated warming estimate reported in Sect. 8 is slightly increased due to the high observed temperatures
in the last few years, which resulted in a further reduction of the budget by around 40 GtCO$_2$, relative to values reported
in last year's assessment (Forster et al. 2024). This gives a total reduction in RCB values estimated from the beginning
of 2025 of ~370 GtCO$_2$ compared to the values from 2020 reported in AR6.

**Table 8 Updated estimates of the remaining carbon budget for 1.5, 1.6, 1.7 and 2.0 °C, for five levels of likelihood,**
**considering only uncertainty in TCRE. Estimates are expressed relative to the start of 2024. The probability includes only**
**the uncertainty in how the Earth immediately responds to CO$_2$ emissions (TCRE), not long-term committed warming or**
**uncertainty in the climate response to other non-CO$_2$ emissions. All values are rounded to the nearest 10 GtCO$_2$. Additional**
**values can be found in the Supplement Tables S7 and S8.**

| Temperature (°C) | Estimated remaining carbon budgets from the beginning of 2025 (GtCO$_2$) | | | | |
|---|---|---|---|---|---|
| Avoidance probability: | 17% | 33% | 50% | 67% | 83% |
| 1.5 | 320 | 200 | 130 | 80 | 30 |
| 1.6 | 620 | 420 | 310 | 240 | 160 |
| 1.7 | 910 | 640 | 490 | 390 | 290 |
| 2.0 | 1790 | 1310 | 1050 | 870 | 690 |


This year's update of the 1.5 °C budget uses the historical warming level for the 2015-2024 period of 1.24 °C, with
0.11 °C future contribution of non-CO$_2$ warming. Assuming a median TCRE estimate of 0.45 °C per 1000 GtCO$_2$ this
gives around 340 GtCO$_2$ from the midpoint of the period, from which we subtract around 210 GtCO$_2$ (204 GtCO$_2$ that
were already emitted from the middle until the end of the 2015-2024 period, and 7 GtCO$_2$ that represents the median





estimate of the impact of Earth systems feedbacks such as permafrost feedback that would otherwise not be covered).
The same method is used to calculate budgets for the other warming levels.
The values in Table 8 are all greater than zero, implying that we have not yet emitted the amount of $CO_2$ that would
commit us to these levels of warming. However, including the uncertainty in ZEC (as in the Supplement Table S8),
non-$CO_2$ emission and forcing uncertainty, and underrepresented Earth-system feedbacks results in negative RCB
estimates for limiting warming to low temperature limits with high likelihood. A negative RCB for a specific
temperature limit would mean that the world is already committed to this amount of warming, and that net negative
emissions would therefore be required to return to the temperature limit after a period of overshoot. The assumption
behind such a calculation is that we can treat the warming impact of positive and negative net emissions as
approximately symmetric. While the claim of symmetry is likely valid for small emissions values, some model studies
have shown that it holds less well for reversal of larger emissions (Canadell et al., 2021, Zickfeld et al., 2021,
Vakilifard et al., 2022, Pelz et al., 2025) As such, larger exceedances of the RCB for a particular temperature target
would decrease the likelihood that the temperature target could still be achieved by an equivalent amount of net
negative emissions.
Note that the 50 % RCB estimate of 130 $GtCO_2$ would be be exhausted in a little more than 3 years if global $CO_2$
emissions remain at 2024 levels (42 $GtCO_2$/yr, see Table 1). This is not expected to correspond exactly to the time
that 1.5 °C global warming level is reached due to uncertainty associated with committed warming from past $CO_2$
emissions (the ZEC) as well as ongoing warming and cooling contributions from non-$CO_2$ emissions. For comparison,
our estimate of 2024 anthropogenic warming (1.36 °C) and the recent rate of increase (0.27 °C/decade) would suggest
that continued emissions at current levels would cause human-induced global warming to reach 1.5°C in
approximately 5 years.
**10 Indicator of climate and weather extremes: land average maximum temperatures**
Changes in climate and weather extremes are among the most visible effects of human-induced climate change. Within
AR6 WGI, a full chapter was dedicated to the assessment of past and projected changes in extremes on continents
(Seneviratne et al., 2021), and the chapter on ocean, cryosphere and sea level changes also provided assessments on
changes in marine heatwaves (Fox-Kemper et al., 2021). Global indicators related to climate extremes include
averaged changes in climate extremes, for example, the mean increase of annual minimum and maximum temperatures
on land (AR6 WGI Chap. 11, Fig. 11.2, Seneviratne et al., 2021) or the area affected by certain types of extremes
(AR6 WGI Chap. 11, Box 11.1, Fig. 1, Seneviratne et al., 2021; Sippel et al., 2015).

The presented climate indicator for changes in temperature extremes consists of land average maximum temperatures
for any single day in a year (TXx) (excluding Antarctica). Fig. 11 updates the land mean TXx shown in Forster et al.
(2023, 2024), originally based on Fig. 11.2 from Seneviratne et al. (2021). Three datasets are analyzed: HadEX3
(Dunn et al., 2020), Berkeley Earth Surface Temperature (building off Rohde et al., 2013), and the fifth-generation



ECMWF atmospheric reanalysis of the global climate (ERA5; Hersbach et al., 2020). HadEX3 is static and has not
received any updates. Berkeley Earth has been extended and updated compared to Forster et al. (2024), resulting in
TXx differences for most years (less than 0.1°C), and now includes data for 2023. Of the three datasets, only ERA5
covers the whole of 2024 at the present time. TXx is calculated by averaging the annual maximum temperature over
all available land grid points (excluding Antarctica) and then converted to anomalies with respect to a base period of
1961–1990. To express the TXx as anomalies with respect to 1850–1900, we add an offset of 0.51 °C to all three
datasets. See Supplement Sect. S10 for details on the data selection, averaging and offset computation.

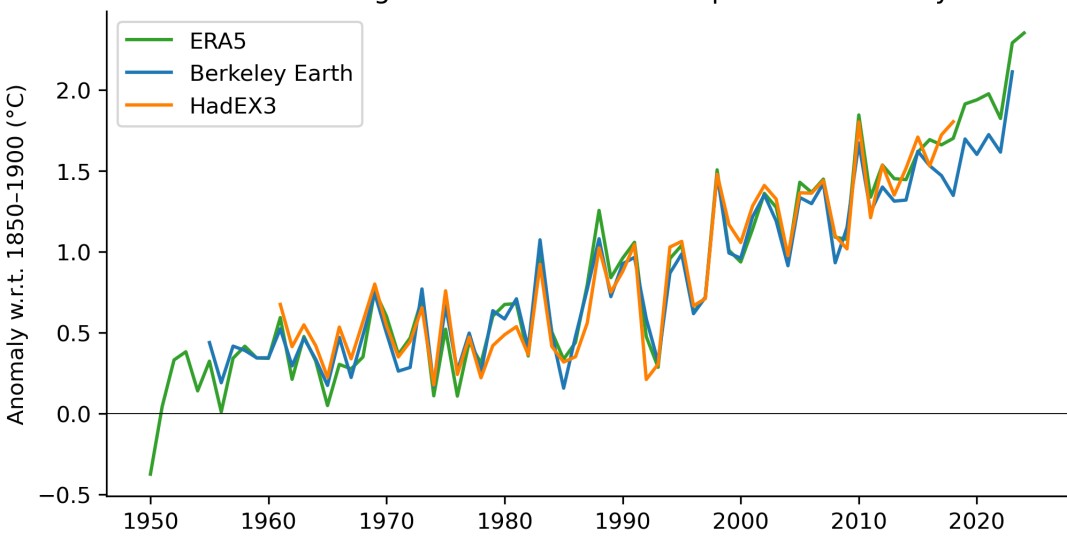


**Figure 11 Time series of observed temperature anomalies for land average annual maximum temperature (TXx) for ERA5 (1950–2024), Berkeley Earth (1955–2023) and HadEX3 (1961–2018), with respect to 1850–1900. The datasets have different spatial coverage and are not coverage-matched. All anomalies are calculated relative to 1961–1990, and an offset of 0.51 °C is added to obtain TXx values relative to 1850–1900. Note that while the HadEX3 numbers are the same as shown in Seneviratne et al. (2021) Fig. 11.2, these numbers were not specifically assessed.**


Our climate has warmed rapidly in the last few decades (Sect. 7), which also manifests in changes in the occurrence
and intensity of climate and weather extremes. From about 1980 onwards, all datasets point to a strong TXx increase,
which coincides with the transition from global dimming, associated with aerosol increases, to brightening, associated
with aerosol decreases (Wild et al., 2005, Sect. 4). The ERA5 based TXx warming estimate w.r.t. 1850–1900 for 2024
is at 2.35 °C; an increase of 0.05 °C compared to 2023, and thus even warmer than the previous record in 2023. On
longer time scales, land average TXx has warmed 0.49 °C in the past 10 years (comparing the decades 2015–2024 to
2005–2014) and 1.90 °C with respect to pre-industrial conditions (Table 9). Since the offset relative to our pre-
industrial baseline period is calculated over 1961–1990, temperature anomalies align by construction over this period
but can diverge afterwards.




**Table 9 Anomalies of land average annual maximum temperature (TXx) for recent decades based on HadEX3, Berkeley Earth, and ERA5, with respect to 1850–1900. All anomalies are calculated relative to 1961–1990, and an offset of 0.51 °C is added to obtain TXx values relative to 1850–1900.**

|  | HadEX3 | Berkeley Earth | ERA5 |
|---|---|---|---|
| 2000–2009 | 1.23 | 1.18 | 1.21 |
| 2005–2014 | 1.37 | 1.31 | 1.4 |
| 2009–2018 | 1.52 | 1.41 | 1.54 |
| 2011–2020 | - | 1.45 | 1.63 |
| 2013–2022 | - | 1.52 | 1.72 |
| 2014–2023 | - | 1.6 | 1.81 |
| 2015–2024 | - | - | 1.9 |


## 11 Global land precipitation

Anthropogenic radiative forcings modify the Earth's energy budget and subsequently drive substantial and widespread
changes in the global water cycle including precipitation, evaporation, atmospheric moisture, and runoff (Forster et
al., 2021, Douville et al., 2021; Gulev et al. 2021). AR6 Chapter 8 assessed that human-caused climate change has
driven detectable changes in the global water cycle since the mid-20th century with high confidence, including an
overall increase in atmospheric moisture (7% per 1 °C of warming), precipitation intensity (1-3% per 1 °C of warming)
and increased terrestrial evapotranspiration (Douville et al., 2021).

In AR6, global land precipitation was highlighted as one of the large-scale indicators of climate change rather than
global precipitation since land precipitation has greater societal relevance and in situ precipitation records over land
extend back to the early to mid-20th century quasi-globally except Antarctica and parts of Africa and South America
(Gulev et al., 2021; Lee et al., 2021; Douville et al., 2021). AR6 assessed that global land precipitation has likely
increased since the middle of the 20th century with a faster increase since the 1980s with large interannual variability
and regional heterogeneity. The observed Northern Hemispheric land summer monsoon precipitation experienced a
significant decline during 1901-2014, which has been attributed to the dominant influence of anthropogenic aerosols
(Cao et al., 2022). Here, we include an update of global land precipitation change since AR6 (i.e., from 2020 to 2024).

Figure 12 shows annual global land precipitation anomaly relative to 1991-2020, following the current WMO
climatology reference, obtained from GPCC V2020 (Schamm et al., 2014), CRU TS 4.08 (Harris et al., 2020), GPCP
V.2.3 (Adler et al., 2018), and GHCN V4 (Menne et al., 2018) observed datasets. There is little consistency among
datasets due to differences in input data, completeness of records, period of covered, and the gridding procedures
applied (Sun et al., 2018; Nogueira, 2020). While the globally averaged land surface specific humidity has
continuously increased (Dunn et al., 2024), global land precipitation has exhibited considerable interannual to
interdecadal variability (Fig, 12). There was a positive anomaly in global land precipitation in 2024 but a negative
anomaly in 2023. The former was contributed to by above-normal precipitation over the Asian and Australian
monsoon region, likely associated with La Nina conditions, but was offset by dry conditions over South America and
the southern part of Africa. The latter was driven by below normal precipitation over South Asia, Maritime Continents,
the southern part of North America and the northern part of South America, due to El Niño conditions, with a
corresponding increase in precipitation over the ocean (Adler and Gu, 2024).

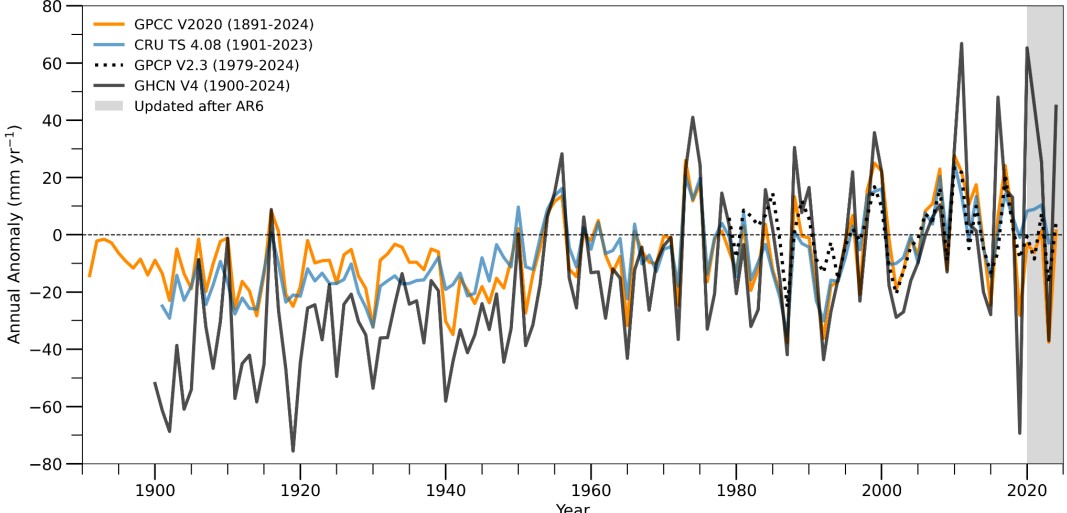


**1030    Figure 12 Time series of annual global land precipitation (mm yr⁻¹) from 1891 to date relative to a 1991-2020 climatology**

**1031    obtained from GPCP V2020, CRU TS 4.08, GPCP V2.3, and GHCN V4 (note that different products commence at distinct**

**1032    times). Annual global land precipitation for each observed data is estimated following the AR6 method except the period of**

**1033    climatology and updated from 2020 to 2024. In AR6, the reference period of the climatology was from 1981 to 2010.**


**1035    12 Global mean sea-level rise**

Global mean sea-level rise (GMSLR) is included in this annual update of AR6 for the first time. GMSLR is primarily
driven by: (i) thermal expansion as the ocean warms; and (ii) increases in ocean mass associated with the addition of
water or ice from land-based reservoirs, including glaciers and ice sheets (Fox-Kemper et al., 2021). Most of these
processes are directly linked to changes in the global Earth energy inventory (Sect. 6). Sea-level rise can have large



consequences for coastal ecosystems, safety and management, as it increases the baseline for sea-level extremes
arising from short-term phenomena such as storm surges, waves and tides.

Observed GMSLR was assessed in IPCC AR6 WG1, in Chapter 2 (their Section 2.3.3.3, Gulev et al., 2021) and
Chapter 9 (their Section 9.6.1 and Cross-Chapter Box 9.1, Fox-Kemper et al., 2021) on the basis of tide gauge
reconstructions (up to 1993) and satellite altimeter observations (1993-2018). The assessment of GMSLR from tide
gauge reconstructions used the ensemble approach presented by Palmer et al. (2021), which quantifies an ensemble
and its uncertainties by combining an estimate of the structural uncertainty (informed by the ensemble spread) with
an estimate of the internal uncertainty across the ensemble (i.e. the parametric uncertainty of each of the members in
the ensemble). The members included in the tide gauge ensemble, which informed the total sea-level change estimate
for the period 1901-1992, were reconstructions from Church and White (2011), Dangendorf et al (2019), Frederikse
et al. (2020) and Hay et al. (2015). For the satellite period, from 1993 to 2018, AR6 used the estimate of the WCRP
Global Sea Level Budget Group (2018), which was constructed from satellite-based GMSLR time series from six
groups (AVISO/CNES, CSIRO, NASA/GSFC, NOAA, SL_cci/ESA and University of Colorado). Based on this
information, AR6 concluded that GMSLR increased by 0.20 [0.15 to 0.25] m over the period 1901 to 2018, with a
rate of 1.73 [1.28 to 2.17] mm yr$^{-1}$ (*high confidence*). Periods closer to the present showed an accelerating GMSLR,
with a rate of 2.3 [1.6 to 3.1] mm yr$^{-1}$ over the period 1971–2018 increasing to 3.7 [3.2 to 4.2] mm yr$^{-1}$ over the period
2006–2018 (*high confidence*).

Here, we extend the AR6 GMSLR time series, which ended in 2018, closer to the present day. We use the same tide
gauge-based ensemble estimate as in AR6 for the period up to 1993. We do note that two new reconstructions have
been published recently, both providing rates in line with the AR6 assessment rates given above. The new GMSLR
reconstruction by Dangendorf et al. (2024) uses a Kalman-smoother and adjusted estimates of the contributions of
glacial isostatic adjustment, barystatic and sterodynamic changes to sea-level change and finds a trend of 1.50 ± 0.20
mm/yr for the period 1900-2021. The new reconstruction by Wang et al. (2024) uses an updated vertical land motion
correction and considers barystatic fingerprints and sterodynamic patterns from CMIP6 models and finds a trend of
1.6 ± 0.2 mm/yr over 1900-2019.

The satellite record now provides observations up to the end of 2024, for three out of the six satellite data products
used for the WCRP estimate used in AR6. The three records available to the end of 2024 are from NASA (2025),
NOAA (2025) and AVISO (2025). All data was downloaded on 19 February 2025. We use the global mean time series
based on the reference missions, with seasonal signals removed and corrected for glacial isostatic adjustment. We first
compute annual averages and then an ensemble average time series, which is spliced to the AR6 GMSLR record
ending in 2018. For consistency, we retain the uncertainties from the six-member WCRP ensemble and propagate
them over the period 2019-2024. We note that reprocessing of the altimetry record is periodically required to account
for new insights on instrument drift, retracking and geophysical corrections to the altimetry missions. This



reprocessing may lead to small differences in the satellite altimeter record and the associated assessment of GMSLR
in future iterations of IGCC.

Over the period 2019 to 2024 global mean sea level has increased by 26.1 [19.8 to 32.4] mm. When combining the
AR6 estimate up to 2018 with the satellite time series for 2019-2024, we find a total GMSLR of 227.0 [176.4 to 229.6]
mm for the period 1901-2024, which translates to an average rate of 1.85 [1.43 to 2.27] mm yr$^{-1}$ (Table 10, Fig. 13).
The rate increase associated with extending the time series by just 6 years, as well as the increasing rates over
consecutive 20-yr periods (Fig. 13b), indicate a continuing acceleration of GMSLR. This is in line with the
assessments of AR6 (Fox-Kemper et al., 2021), SROCC (Oppenheimer et al., 2019) and AR5 (Church et al., 2013)
that sea-level change has been accelerating over course of the 20th and early 21st centuries, and consistent with the
observed acceleration in some components of the Earth heat inventory (see Sect. 6).

**Table 10 Observed global mean sea-level rise (GMSLR) as presented in IPCC AR6, table 9.5 (Fox-Kemper et al., 2021)**
**compared with the extended time series in this study. Values are expressed as the total change (Δ) in the annual mean**
**over each period (mm) along with the equivalent rate calculated as the total change divided by the number of years (mm**
**yr$^{-1}$). Uncertainties represent the *very likely* range.**

| Observed GMSLR | | IPCC AR6 | This study |
|---|---|---|---|
| **Start year** | | **End year 2018** | **End year 2024** |
| **1901** | Δ(mm) | 201.9 [150.3 to 253.5] | 227.0 [176.4 to 229.6] |
| | mm yr$^{-1}$ | 1.73 [1.28 to 2.17] | 1.85 [1.43 to 2.27] |
| **1971** | Δ(mm) | 109.6 [72.8 to 146.4] | 135.8 [99.0 to 172.5] |
| | mm yr$^{-1}$ | 2.33 [1.55 to 3.12] | 2.56 [1.87 to 3.26] |
| **1993** | Δ(mm) | 81.2 [72.1 to 90.2] | 107.3 [98.2 to 116.4] |
| | mm yr$^{-1}$ | 3.25 [2.88 to 3.61] | 3.46 [3.17 to 3.75] |
| **2006** | Δ(mm) | 44.3 [38.6 to 50.0] | 70.4 [64.7 to 76.1] |
| | mm yr$^{-1}$ | 3.69 [3.21 to 4.17] | 3.91 [3.59 to 4.23] |





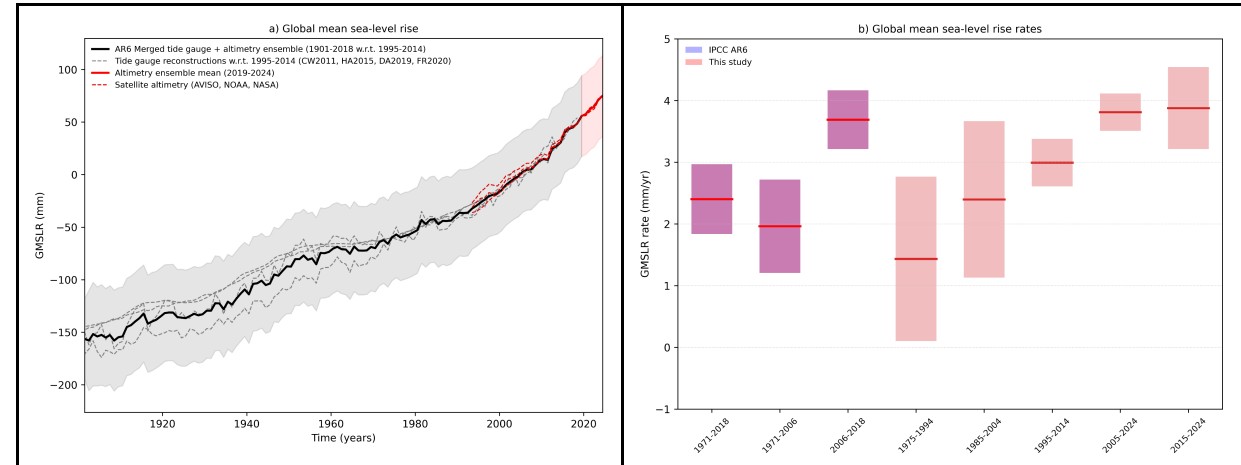

**Figure 13 (a) Global mean sea-level rise time series 1901-2024 (mm). The GMSLR ensemble from AR6 in black, w.r.t. the period 1995-2014; the updated satellite altimetry ensemble in red, w.r.t. the AR6 ensemble in 2018. Individual time series are shown in dashed lines. (b) GMSLR rates (mm yr-1) for different periods. Uncertainties in a) show the *likely* range and in b) the *very likely* range, computed relative to 1901, including estimates of both structural uncertainty and parametric uncertainty (Palmer et al., 2021).**

**13 Code, data availability and visualisations**

We publish a set of selected key indicators of global climate change via Climate Change Tracker (https://climatechangetracker.org/, Climate Change Tracker, 2025), a platform which aims to provide reliable, user-friendly, high-quality interactive dashboards, visualisations, data, and easily accessible insights of this paper.

With Climate Change Tracker we aim to reach a wider public audience, including policymakers involved in UNFCCC negotiations, and decision makers working in climate change mitigation and adaptation. Climate Change Tracker plans to update significant indicators multiple times throughout the year, providing an up-to-date picture of the indicators of climate change. Within the dashboards, all data is traceable to the underlying sources.

The carbon budget calculation is available from https://github.com/Rlamboll/AR6CarbonBudgetCalc/tree/v1.0.3 (Lamboll and Rogelj, 2025). The code and data used to produce other indicators are available in repositories under https://github.com/ClimateIndicator/data/releases/tag/v2025.04.30b (Smith et al., 2025b). All data are available from https://doi.org/10.5281/zenodo.15327155 (Smith et al., 2025a). Data are provided under the CC-BY 4.0 License.

HadEX3 [3.0.4] data were obtained from https://catalogue.ceda.ac.uk/uuid/115d5e4ebf7148ec941423ec86fa9f26 (Dunn et al., 2023) on 5 April 2023 and are © British Crown Copyright, Met Office, 2022, provided under an Open



Government Licence; http://www.nationalarchives.gov.uk/doc/open-government-licence/version/2/ (last access: 2
June 2023).
**14 Discussion and conclusions**
The third year of the Indicators of Global Climate Change (IGCC) initiative has built on previous years' efforts to
provide a comprehensive update of the climate change indicators required to estimate the human-induced warming
and the remaining carbon budget. Table 11 and Fig. 14 present a summary of the headline indicators from each section
compared to those given in the AR6 assessment.  Table 11 also summarises methodological updates.
**Table 11 Summary of headline results and methodological updates from the Indicators of Global Climate Change (IGCC)**
**initiative.**

| Climate Indicator | AR6 2021 assessment | This 2024 assessment | Explanation of changes | Methodological updates since AR6 |
|---|---|---|---|---|
| **GHG emissions** <br><br> **AR6 WGIII Chapter 2: Dhakal et al. (2022); see also Minx et al. (2021)** | **2010-2019 average:** <br><br> **55.9 ± 6 GtCO₂e** | **2010-2019 average:** <br><br> **52.9 ± 5.4 GtCO₂e** <br><br> **2014-2023 average:** <br><br> **53.6 ± 5.2 GtCO₂e** | **Average emissions in the past decade grew at a slower rate than in the previous decade. The change from AR6 is due to a systematic downward revision in CO₂-LULUCF and CH₄ estimates. Real-world emissions have slightly increased.** | **CO₂-LULUCF emissions revised down. CO₂ GCB Fossil Fuel and Industry emissions used instead of EDGAR. PRIMAP-hist TP used in place of EDGAR for CH₄ and N₂O emissions, atmospheric measurements taken for F-gas emissions. These changes reduce estimates by around 3 GtCO₂e (Sect. 2).** |
| **GHG concentrations** <br><br> **AR6 WGI Chapter 2: Gulev et al. (2021)** | **2019:** <br><br> **CO₂, 410.1 [± 0.36] ppm** <br><br> **CH₄, 1866.3 [± 3.2] ppb** <br><br> **N₂O, 332.1 [± 0.7] ppb** | **2024:** <br><br> **CO₂, 422.8 [±0.4] ppm** <br><br> **CH₄, 1929.8 [±3.3] ppb** <br><br> **N₂O, 337.9 [±0.4] ppb** | **Increases caused by continued GHG anthropogenic emissions** | **Updates based on NOAA data and AGAGE (Sect. 3)** |



| | | | | |
|---|---|---|---|---|
| **Effective radiative forcing change since 1750**<br><br>**AR6 WGI Chapter 7: Forster et al. (2021)** | **2019:**<br><br>**2.72 [1.96 to 3.48] W m$^{-2}$** | **2023:**<br><br>**2.97 [2.05 to 3.76] W m$^{-2}$** | **Trend since 2019 is caused by increases in GHG concentrations and reductions in aerosol precursors.** | **Follows AR6 with minor update to aerosol precursor treatment and emissions dataset that revises 2019 ERF estimate relative to 1750 downwards (more negative) by 0.09 W m$^{-2}$. Added this year is a new method to estimate the ERF from land use surface reflection and irrigation to avoid scaling with cumulative emissions. This does not materially affect the ERF. (Sect. 5)** |
| **Earth's energy imbalance**<br><br>**AR6 WGI Chapter 7: Forster et al. (2021)** | **2006-2018 average:**<br><br>**0.79 [0.52 to 1.06] W m$^{-2}$** | **2012-2024 average:**<br><br>**0.99 [0.70 to 1.28] W m$^{-2}$** | **A 25% increase in energy imbalance estimated based on increased rate of ocean heating.** | **Ocean heat content timeseries extended from 2018 to 2024 using all of the 5 AR6 datasets. Other heat inventory terms updated following von Schuckmann et al. (2023a). Ocean heat content uncertainty is used as a proxy for total uncertainty. Further details in Sect. 6.** |
| **Global mean surface temperature change since 1850-1900**<br><br>**AR6 WGI Chapter 2: Gulev et al. (2021)** | **2011-2020 average:**<br><br>**1.09 [0.95 to 1.20] °C** | **2014-2023 average:**<br><br>**1.24 [1.11 to 1.35] °C** | **An increase of 0.15 °C within four years, indicating a high decadal rate of change which may in part be internal variability.** | **Methods match four datasets used in AR6. Individual datasets have updated historical data, but these changes are not materially affecting results. (Sect. 7).** |
| **Human induced global warming since preindustrial**<br><br>**AR6 WGI Chapter 3: Eyring et al. (2021)**<br><br><br>**SR1.5 Chapter 1** | **2010-2019 decade average:**<br><br>**1.07 [0.8 to 1.3] °C**<br><br>**2017 single year: 1.0 [0.8 to 1.2] °C** | **2015-2024 decade average:**<br><br>**1.22 [1.0 to 1.5] °C**<br><br>**2024 single year: 1.36 [1.1 to 1.7] °C** | **An increase of 0.15 °C within five years, indicating a high decadal rate of change (broadly consistent with warming projections). The decadal warming rate increased slightly between 2019 and 2024. One of the three AR6 methods is diverging.** | **The three methods for the basis of the AR6 assessment are retained, but each has new input data (Sect. 8)** |



| | | | |
|---|---|---|---|
| **Remaining carbon budget for 50% likelihood of limiting global warming to 1.5 °C**<br><br>**AR6 WGI Chapter 5: Canadell et al. (2021)** | **From the start of 2020:**<br><br>**500 GtCO$_2$** | **From the start of 2025:**<br><br>**130 GtCO$_2$** | **The 1.5 °C budget is becoming very small. The RCB can exhaust before the 1.5 °C threshold is reached due to having to allow for future non-CO$_2$ warming.** | **Emulator and scenario change has reduced budget since 2020 by 100 GtCO$_2$ (Sect. 9)** |
| **Land average maximum temperature change compared to pre-industrial.**<br><br>**AR6 WGI Chapter 11: Seneviratne et al., 2021** | **2009-2018 average:**<br><br>**1.55 °C** | **2015-2024 average:**<br><br>**1.90 °C** | **Rising at a substantially faster rate compared to global mean surface temperature** | **HadEX3 data used in AR6 replaced with ERA reanalysis data employed in this report which is more updatable going forward. Adds 0.01 °C to estimate (Sect. 10)** |
| **Global land precipitation compared to preindustrial (Douville et al., 2021)** | **Likely increased since the middle of the 20th century with a faster increase since the 1980s with large interannual variability** | **Large interannual variability associated with El Niño dominates the record in recent years, making long-term trend less clear** | **2023 exhibited a negative anomaly relative to preindustrial due El Niño conditions** | **The four datasets used in AR6 have been extended (Sect. 11)** |
| **Global mean sea-level rise since 1901**<br><br>**(Gulev et al., 2021; Fox-Kemper et al., 2021)** | **1901 to 2018 change**<br><br>**201.9 [150.3 to 253.5] mm**<br><br>**at a rate of**<br><br>**1.73 [1.28 to 2.17] mm yr$^{-1}$** | **1901 to 2024 change**<br><br>**227.0 [176.4 to 229.6] mm**<br><br>**at a rate of**<br><br>**1.85 [1.43 to 2.27] mm yr$^{-1}$** | **Sea-level rise continues to accelerate.** | **AR6 data extended with three of the six datasets from AR6, using latest satellite data (Sect. 12).** |








**Figure 14 Infographic for the best estimate of headline indicators assessed in this paper.**
Last year (2024) witnessed global surface temperatures likely exceeding 1.5 °C above preindustrial levels which has
widely been reported in the press. Sects. 7 and 8. show that such high levels of global temperature anomalies are
typical of what we expect from current best estimates of human induced warming, modulated by internal climate
variability.

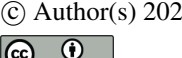



The overview of key indicators of the state of global climate indeed highlights the multiple fingerprints of the 2023-
2024 El-Nino event regarding peak global surface temperature (Section 7.2), regional dry anomalies in land
precipitation (Section 11), and their implications for reduced land carbon sinks and the record growth rate of
atmospheric $CO_2$ concentrations in 2024 (Section 3).
The overall increase in land maximum temperatures (Section 10), closely related to global warming levels, drives
increasing trends in potential evapotranspiration, decreasing trends in soil moisture (Seo et al., 2025), contributing to
the increased rate of global mean sea-level rise (Section 12).
Methane and biomass emissions had a strong component of change related to climate feedbacks (Sects. 2 and 3). Such
changes will become increasingly important over this century, even if the direct human influence declines. This year,
we explored different inventory choices in Sect. 2. In future years a more consistent approach to attribution of
atmospheric emissions, concentration change and radiative forcing should be developed, so it can be assessed in AR7.
It is hoped that this update can support the science community in its collection and provision of reliable and timely
global climate data. In future years we are particularly interested in improving SLCF updating methods to get a more
accurate estimate of short-term ERF changes. The work also highlights the importance of high-quality metadata to
document changes in methodological approaches over time. This year we have extended the datasets with land
precipitation and global mean sealevel rise. In future years we hope to improve the robustness of the indicators
presented here and could update other AR6 assessments. Parallel efforts could explore how we might update indicators
of regional climate extremes and their attribution, which are particularly relevant for supporting actions on adaptation
and loss and damage.

Generally, scientists and scientific organisations have an important role as "watchdogs" to critically inform evidence-
based decision-making. This annual update traced to IPCC methods can provide a reliable, timely source of
trustworthy information. It very much relies on continued support for high quality global monitoring networks of
atmospheric and climate data, and also on open data sources that are regularly updated and easily accessed.

This is a critical decade: human-induced global warming rates are at their highest historical level, and 1.5 °C global
warming might be expected to be reached or exceeded in around 5 years in the absence of cooling from major volcanic
eruptions (Sects. 8 and 9). Yet this is also the decade when global GHG emissions could be expected to peak and
begin to substantially decline. The indicators of global climate change presented here show that the Earth's energy
imbalance has increased to around $1.0 \, \text{W m}^{-2}$, averaged over the last 12 years (Sect. 5), which represents a 25%
increase on the value assessed for 2006-2018 by AR6. This also has implications for the committed response of slow
components in the climate system (glaciers, deep ocean, ice sheets) and committed long-term sea-level rise (through
ocean thermal expansion and land-based ice melt/loss), to be addressed further in future updates. However, rapid and
stringent GHG emission decreases such as those committed to at COP28 could halve warming rates over the next 20





years (McKenna et al., 2021). Table 1 shows that global GHG emissions are at a long-term high, yet there are signs
that their rate of increase has slowed. Depending on the societal choices made in this critical decade, a continued series
of these annual updates could track an improving trend for some of the indicators herein discussed.
**Supplement**
The supplement related to this article is available online.
**Author contributions**
PMF, CS, MA, PF, JR and AP developed the concept of an annual update in discussions with the wider IPCC
community over many years. CS led the work of the data repositories. VMD, PZ, SS, CS, SIS, VN, AP, NPG, GPP,
BT, MDP, KvS, JR, PF, MA, JCM, XZ, RAB, CB, CC, SB and PT provided important IPCC and UNFCCC framing.
PMF coordinated the production of the manuscript with support from DR. WFL led Sect. 2 with contributions from
PF, GPP, JG, JP, JCM and RA. CS led Sect. 3 with inputs from JM, PK, LW, PMF and MR. SS led Sect. 4 with inputs
from VK, CS, GvdW, LAR and MG. CS led Sect. 5 with contributions from CW, TG, SS, VN and GvdW. KvS and
MDP led Sect. 6 with contributions from LC, MI, JR, REK, AS, CMD, DPM and SEW. BT, CC and ZH led Sect. 7
with contributions from PT, CM, CK, JK, RR, RV, AL and LC. TW led Sect. 8 with contributions and calculations
from AR, NPG, SJ, CS and MA. RL led Sect. 9 with contributions from JR and HDM. Sect. 10 was led by MH, with
contributions from SIS, and XZ. JYL, JEY, and RK led Sect. 11 with contributions from VMD, PT, and KvS. AS led
Sect. 12 with contributions from MDP. All authors either edited or commented on the manuscript. DR, AB and JAB
coordinated the data visualisation effort.
**Competing interests**
The contact author has declared that none of the authors has any competing interests.
**Disclaimer**
Publisher's note: Copernicus Publications remains neutral with regard to jurisdictional claims in published maps and
institutional affiliations.
**Acknowledgements**
This research has been supported by the European Union's Horizon Europe research and innovation programme under
Grant Agreement Nos. 820829, 101081395, 101081661 and 821003, the H2020 European Research Council (grant
no. 951542), the Natural Environment Research Council (NE/X00452X/1) and the Engineering and Physical Research
Council (EP/V000772/1). Matthew Palmer, Colin Morice, Rachel Killick and Richard Betts were supported by the
Met Office Hadley Centre Climate Programme funded by DSIT. Peter Thorne was supported by Co-Centre award
number 22/CC/11103. The Co-Centre award is managed by Research Ireland Northern Ireland's Department of



Agriculture, Environment and Rural Affairs (DAERA) and UK Research and Innovation (UKRI), and supported via
UK's International Science Partnerships Fund (ISPF), and the Irish Government's Shared Island initiative. Analyses
and visualizations for concentrations of Short Lived Climate Forcers used in this paper were produced with the
Giovanni online data system, developed and maintained by the NASA GES DISC (as available in February
2025). June-Yi Lee and Jung-Eun Yun were supported by the National Research Foundation of Korea (NRF)
grant funded by the Korea government (MSIT) (No. RS-2024-00416848). Aimée Slangen was supported by the
research programme ENW-Vidi (DARSea, project number VI.Vidi.2023.058) funded by the Dutch Research Council
(NWO). We thank Xin Lan for assistance with compiling the GHG concentration data.

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
