# Peer review of "Indicators of Global Climate Change 2024: annual update of key"

_Earth System Science Data, 2025_

## Author Comment (AC1)

**Response to referee comments and public comments**

Note that in the below text referee and public comments are given in italics

**Response to RC1, John Dunne**

We thank John very much for his incredibly helpful review comments. We agree with his assessment and have tried to be more careful with the treatment of uncertainty, especially around human induced warming in our response and manuscript revision. Detailed responses are given below.

*"My main scientific criticism is in the interpretation that the authors are able to attribute all 1.19°C of the 1.19°C of observed decadal warming as human-induced as there are some questionable implications of these central values being the same, first the idea that the authors can discount any role of decadal to centennial scale climate variability of potentially playing any significant role, and second that the alignment of these central values seems to imply that the human-induced warming response of the real world was just as likely to be more severe as less severe than what has occurred. This seems more likely a flaw in the approach in ignoring potential sources of uncertainty than a statement of absolute alignment between effective radiative forcing and global response."*

We broadly agree and have tried to be more careful to caveat the uncertainties over the human induced warming. It is perhaps worth noting that the 1.19°C results quoted in this comment are those from the 2023 Indicators paper (published in 2024), and that in this year's 2024 Indicators paper (under discussion here) the central estimates for decadal observed warming and human induced warming do not exactly match for the first time (1.24°C and 1.22°C respectively). However, the broader point about these values' close alignment remains relevant. For context, the two values are calculated using separate analyses; the central estimate for attributed anthropogenic warming in particular is given as the multi-method mean of three attribution methods, each of which has slightly different central estimates and uncertainty ranges; the very conservative uncertainty estimate reflects these inter-method differences. The framing we use (last paragraph of §8.2) is that, within the uncertainty ranges, the observed warming is very similar to the attributable human-induced warming, which we believe is a reasonable interpretation of the results given the uncertainty ranges involved and various methodological limitations. We have not made the claim of explicitly discounting any role for climate variability in the headline results; this specific role of internal variability has been more thoroughly covered for the 2024 single-year warming in §7.2. The attribution assessment follows the IPCC assessments as closely as possible, and we do not want to change (it is beyond the scope of this initiative). It is worth noting that the attribution methods do somewhat account for internal variability (e.g. in the case of the GWI method, O(100) samples of possible internal variability realisations are used); the role of internal variability is therefore incorporated into the uncertainty

range for the attributed warming results. We also speculate, as the referee suggests, that there could be limitations with the attribution methods that might be picked up in the next IPCC report; some of these are outlined in the supplement. We think the abstract was pretty clear over the different ranges, but have edited the text in several places to improve clarity.

*"My main communication criticism of the current manuscript is the lack of reference to/contextualization in the introduction and motivation for the present work against the two other community level annual reports offered annually on this topic, namely the BAMS "State of the Climate" and WMO "State of the Global Climate" reports put out every year. The big difference I see is that the present analysis goes beyond the observations into process level effective radiative forcing and attributed human-induced response using the same (at times updated) methods rigorously assessed in AR6. This oversight is particularly odd given that both products are referenced later in the manuscript, so the authors are obviously aware of them and use them as sources in their analysis. "*

Statements on these WMO and BAMS comparisons were removed for brevity as they repeated previous years text. They have been reinstated to improve the contextualisation. We borrowed some of your helpful wording suggestions, thank you!

*Abstract: remove second "on the global climate system" in second sentence.*
This text did not seem to exist in the document pdf version?

*Abstract: remove "(AR6)" and (WGI)" as these acronyms are not used in the abstract.*
AR6 is used but WGI is deleted

*Abstract: remove or rephrase "can be trusted by all parties involved" – the question of "trust" is separate from whether the research result is a consistent extension of previously trusted methods as a new research result.*

This text was not in our abstract. I think there must be some confusion? John appears to have reposted his 2024 paper review?

*Abstract: remove "its direction of travel"*

This text was not in our abstract. I think there must be some confusion, see above.

Introduction, first paragraph: "is there any other provenance to the IGCC activity such as endorsement or sponsorship by WCRP, IPCC, specific government initiatives etc? Funding sources?"

As IGCC started as a bottom up initiative without funding, and as it is not officially "endorsed" or supported by particular governments, we prefer not to add further text.

*Introduction, first paragraph: Here is where a brief statement on why this IGCC effort is necessary given that BAMS already puts out the "State of the Climate" and WMO "State of the Global Climate" every year. How is this different/complementary? The use of the same methods as in AR6? Is it more comprehensive of forcings, model information and budgets in this way compared to BAMSWMO focus only on observations? See above for suggested language.*
Text added to introduction using your helpful wording, thank you.

*Introduction, second paragraph: "last year" should specify that the focus of Forster et al., 2023 was an extension of the methods through 2022.*
Text added for clarity

Introduction, second paragraph: "the remaining carbon budget" is only relevant when specified for particular climate warming thresholds, so should indicate the remaining carbon budget for what? 1.5C? 2C? 3C?... add "to policy-relevant temperature thresholds".

Text added on 1.5C and policy-relevant thresholds

*Introduction, third paragraph: The statement that the AR6 methods are not being used exactly but are including "evolving methodological improvements" seems to conflict with the objective of using the same methods as AR6 to be "trusted"… what if one person's "methodological improvement" is another person's wayward foray? Who will be the judge that UNFCCC parties should "trust" this new method as much or more than those that were vetted through the AR6 process?*

There is a certain amount of author judgment here, but as far as possible AR6 methods are retained. We hope the current text along with the details in the sections and table 11, give readers enough detail of any methodological modifications. We quantify and the give the effect of these changes in the text and so far they are very small.

*Introduction, fourth paragraph: Suggest replacing "The update" with "This annual update"*
Text changed as suggested

*Treatment of wildfires – The statement "The GCB methodology includes CO2 emissions from deforestation and forest degradation fires but excludes wildfires, which are assumed to be natural even if climate change affects their intensity and frequency" should also note that these fires are treated as "natural" even when they are ignited by human activity to clarify that many "wildfires" are due to human activities. Also, it is not clear how fires ignited purposely for ongoing land use on croplands (e.g. sugarcane) are treated?*

Fire treatment is complex, we have clarified text used from the previous years.

We appreciate that the emissions reporting convention is in some cases subjective and depends on the accounting method; several of the author team address this issue in a separate submitted manuscript (Lamb et al. 2025). Following the spirit of IGCC, we have attempted to follow conventions chosen in the IPCC AR6 WGIII as closely as possible. While far from a perfect separation, the fire emissions from deforestation and forest degradation are treated as a medium to long term net transfer of carbon from the biosphere to the atmosphere, and primarily (though not exclusively) the result of human activities. The fire emissions from wildfires are treated as "short run", and there is an implicit assumption that burned forest will recover and draw down the emitted carbon, though of course this process could take decades. Moreover, while there is a human component to wildfires (agricultural burning, accidents, arson, etc), there is a natural component, and furthermore the natural component is changing due to climate change (Burton et al., 2024). The difficulty arises as there does not exist a robust method to cleanly separate anthropogenic and natural components of fires, and we comment on this complexity in the section 4 on short-lived climate forcers: "Fires can be worsened by climate change, because of increased fire prone weather conditions (Burton et al., 2024). Strictly speaking, such fires could sometimes be considered as feedbacks and not be included in anthropogenic forcings. However, we choose to include fires in our tracking, as historical biomass burning emissions inventories have previously been consistently treated as an anthropogenic forcing (for example in CMIP6), though this assumption may need to be revisited in the future (see also discussion in Sect. 5).

Lamb, W., Andrew, R., Jones, M., Nicholls, Z., Peters, G., Smith, C., Saunois, M., Grassi, G., Pongratz, J., Smith, S., Tubiello, F., Crippa, M., Gidden, M., Friedlingstein, P., Minx, J., and Forster, P.: Differences in anthropogenic greenhouse gas emissions estimates explained, https://doi.org/10.5194/essd-2025-188, 24 April 2025.

Burton, C., Lampe, S., Kelley, D. I., Thiery, W., Hantson, S., Christidis, N., Gudmundsson, L., Forrest, M., Burke, E., Chang, J., Huang, H., Ito, A., Kou-Giesbrecht, S., Lasslop, G., Li, W., Nieradzik, L., Li, F., Chen, Y., Randerson, J., Reyer, C. P. O., and Mengel, M.: Global burned area increasingly explained by climate change, Nat. Clim. Chang., 14, 1186–1192, https://doi.org/10.1038/s41558-024-02140-w, 2024.

*Table 1: Should note that "GHG" in row one signifies the total of the other rows.*
Text added to caption to note this.

*Table 2: Given that none of the numbers for 2019 and 2022 are exactly the same as in Forster et al., 2023 – It should be noted in the caption that they are all updated.*

Noted, as suggested

*Section 4: I am not sure I agree with the statement that wildfires are a "climate feedback" in "it is not easy to determine how much of the biomass burning contribution is from natural wildfires in response to 2023's anomalously warm year, which would be a climate feedback rather than a forcing." In cases where the fires were ignited by human activity. Certainly, "it is not easy", but I think it is even harder than the authors indicate.*

The statement on feedback has been caveated by adding " could sometimes be". See also response to CO2 fires above.

*Section 11: The statement "This annual update traced to IPCC methods can provide a reliable, timely source of trustworthy information." Should add something like "joins the State of the Climate (BAMS) and State of Global Climate (WMO) reports"*

Text added line line with suggestion

**Response to PC1, Gareth Jones**

Gareth Jones is thanked yet again for another comprehensive review of our work. We really welcome it and the rigour it applies. We have thought about the points made carefully and have tried to be as clear as possible over data. In our response we have added further clarity on datasets and uncertainty. On differences of opinion, we have tried to present main differences in the text, but would prefer to leave a full assessment of the literature for the next IPCC report.

*The Indicators of Global Climate Change reports are ambitious. However, to be of value to "evidence-based decision-making", they need to be more robust about presentation of uncertainties, dataset choice, and differences of opinion.*

We have specified dataset choices carefully and shown more differences compared to previous years, e.g. with emission and precipitation datasets. We add further details on uncertainty in the revision. On differences of opinion, we have tried to present main differences in the text, but would prefer to leave a full assessment of the literature for the next IPCC report.

*This study would be much more useful for other scientists and policy-makers if it did not try to link itself to the IPCC assessment cycles so closely.*
*The studies have gone through much less review than IPCC reports have.*
*While the authors have tried to follow FAIR principles - like the IPCC - there are too many cases of data and code not being publicly accessible.It seems arbitrary which IPCC methods are followed "as closely as possible" and which are modified or completely new. Inconsistencies between the IGCC studies for results of the same periods make it difficult for anyone to "Track ... between IPCC report cycles". Many differences in "updates" for latest year/decade will be due to changes in method and data source, and may not be as representative of climate change as is often suggested.*

*AR7 could be hindered if there is an attempt to force a continuation of a later IGCC paper with it.*
*Using terms like "WGIII update" & "IPCC update" give impression this is an official IPCC update, when it is not.*

We disagree with these points. We think that the methods can be traced back to IPCC is important and that any departures are expressly noted and their effects quantified in the text, supplementary material and summarised in Table 11. Also the code used directly derives from that used in AR6. We think the Kadow dataset is the one referred to as not publicly available. We use it as it was adopted by IPCC. In response we now publish the latest dataset online at Kadow, C., Plésiat, É., & Lenssen, N. (2025). Annual update of Climate Reconstruction AI (CRAI) infilled HadCRUT5 of near-surface temperature change 1850 to 2024 [Data set]. Zenodo. [https://doi.org/10.5281/zenodo.15622091](https://doi.org/10.5281/zenodo.15622091). This year we have further added further comparison to WMO state of the Global Climate choices. We rather think the effort benefits AR7, especially as the AR7 outline expressly requests that authors assess annual indicators and their methods. We respond to your points in turn below.

*General comments*
* * *
*\* FAIR principles:  Not all code for producing the figures is available. The github and zenodo links are somewhat confusing with discrepancies with what is in Section 13 and what is in the references. Dataset version numbers are not always given. There are no direct links to datasets within the manuscripts. It looks like the authors expect readers to hunt throughout the github/zenodo links to find the links. These links are not easily found ... if at all. There are mistakes in some of the repositories, which have been there since F23.*

The Zenodo repository is entirely new and is a snapshot of the github pages. We think it is better to present all the data together with one link and matching github page, rather than confuse the reader with lots of different references. This helps IGCC be a stand alone project. Code for the missing figures has been added, Figures 4, 7 and 8.

*There seems to be an increase in the number of values being presented without any uncertainties, over what was in F23 and F24.*

For clarity of writing uncertainties are not presented in every instance, but each section discusses uncertainties and they are in the headline figures of abstract and Table 11. Best estimates are also discussed where we think it is appropriate to consider these.

*For instance, no uncertainties are given for the observed temperatures for 2024 anywhere in the paper, despite it being a prominent result.*

This is due to the way the homogenized dataset for surface temperature is constructed - a range of temperatures is now given in the text. Section 7.2 discusses these uncertainties at length.

*Various statements are not supported when consideration of the different sources of uncertainty are accounted for.*

We have carefully checked the main statements and argue they are justified by the data and the analysis.

*How can one interpret differences in results for the latest year/decade between reports, when values for the same periods change from report to report? e.g.,*

*1) First five columns of numbers in Table 1 are different to values for the same periods in Table 1 in F23, Table 1 in F24 and with the rows of three of the periods in Table 2.1 in IPCC AR6 WGIII.*

We have added an explanation to the text that recent emission estimates are uncertain and they are revised as the underlying datasets are updated. These small changes to values are a routine part of the dataset production process. Table S1 shows the exact datasets employed.

*2) Values in Table 2 for "2019 (updated)" differ from those for "2019" in table 2 in F23 and table 2 in F24*

Again, this is due to routine revisions to the underlying datasets as discussed in the text. We have added an explanation to the caption. Section S4 and Table S3 presents a direct comparison with the past estimates from our earlier papers.

*3) Values in Table 6 for "repeat calculations" for 2010-2019 and 2017 differ from the "repeat calculations in F23 (table 6) and F24 (Table 6)*

This is correct, and due to the calibration of the temperature attribution methods to the latest temperature and forcing data as explained in the text; we re-assess the the same historical

period used in SR1.5 and AR6 in each year's update in order to identify how much of the year-on-year change is due to (a) actual forced changes in the climate between 2023 and 2024, and (b) simply a revision of the historical period from re-running the attribution methods with the updated datasets and methods. Indeed, this reflects this reviewer's earlier comment: "Many differences in "updates" for latest year/decade will be due to changes in method and data source, and may not be as representative of climate change as is often suggested." A granular discsussion about this previously discussed in depth in the main text (see the final paragraph on page 2642 of F24).

*4) Table S2 has subtly different concentrations, for same years in F23 (table S1) and F24 (Table S2)*

As already explained in the text this is due to very small changes to the calibrations within the underlying data.

*\* New science*

*In various places new analyses are presented. These really should be presented in separate studies, to allow a full peer-review assessment to take place.*

*e.g., the analysis in Figure 8, the method to estimate volcanic forcing in 2024 (SI S5.5.2), or the analysis in S7 "2023 and 2024 temperature anomalies".*

Figure 8 is based on analysis following Chapter 3 of AR6 Working Group 1 (Erying et al., 2021). This discussion was felt appropriate given the large warming anomalies for 2023-2024. The original IPCC method of estimating volcanic forcing was not appropriate for 2023-2024 conditions so as detailed in the supplement, we took advantage of satellite data to derive an estimate. This is detailed in the supplement.

*There also appears to be a lack of consistency with assumptions, choices and data used in some of these new analyses differing to what is used in the rest of the paper.*

We think this refers to the Hausfather attribution analysis in Figure S6 of the supplement. This analysis is added for further discussion on recent warming, it is presented to be illustrative and does not affect the data or conclusions of the paper. We think it adds useful context and discussion.

*\* New volcanic forcing*

*The volcanic forcing in Figure 5 looks quite different to figure 3 in F24. It is due to a different volcanic forcing dataset being used. The proposed CMIP7 dataset has not yet been peer reviewed, and given the very different methods to construct it, there are questions which*

*really need to be addressed before it gets used in anger. e.g., The aerosol distribution following the Mt Agung eruption was robustly observed to be mostly in the Southern Hemisphere, but the dataset has a pretty much hemispheric symmetrical distribution.*

We argue that the transition from CMIP6 to CMIP7 (volcanic forcing dataset v1.3.0 used here) as the basis for the 1850 to "recent past" portion of the stratospheric aerosol optical depth used in the volcanic ERF calculation is justified on the IGCC basis of using updated datasets. CMIP7 (v1.3.0) includes a better representation of small eruptions and a more recent calibration of the volcanic aerosol model, as well as including additional years. Up until F24 we were using our own methods to extend the CMIP6 time series (ending in 2014) up until the present day, so it is better to have something more official up until 2023 where the extensions that we make are for a shorter time period. We are aware of a forthcoming version (v2.2.1) of the CMIP7 volcanic forcing that fixes the Agung inter-hemisperic issue.

*\* Aliasing*

*When comparing results for adjacent overlapping periods it is harder to interpret the results than the authors suggest, due to aliasing. The significance of such changes should be treated cautiously.*

*e.g., In Section 7.1 and table 5, interannual variability projects onto the differences between overlapping 10 year means.*

*The reported 0.15C increase between 2011-2020 and 2015-2024 is actually half of the difference between 2011-2014 and 2021-2024, so has more "noise" than differences between non-overlapping 10 year means.*

These numbers are presented to illustrate changes since the last IPCC report and show consistency with more detailed and rigours warming rate trend estimates given later. We think the comparisons are useful.

*\* Kadow GMST observations*

*It is astonishing that an observational temperature dataset is still being used that has not been documented (there is no paper/report describing the Kadow model using the HadCRUT5 observational dataset), and is not publicly accessible.*

We adopt Kadow et al. as it was used in AR6 and we follow their methods - we now also add the dataset online ar Kadow, C., Plésiat, É., & Lenssen, N. (2025). Annual update of Climate Reconstruction AI (CRAI) infilled HadCRUT5 of near-surface temperature change

1850 to 2024 [Data set]. Zenodo. https://doi.org/10.5281/zenodo.15622091 and reference this in the supplement.

*\* Human-induced warming attribution*

*I continue to be troubled by the lack of a presentation of the strength/weaknesses of the different methods.*

Thank you for your detailed review and comments on this.

As stated throughout this paper and in the previous iterations of this initiative, the scope of this work is to adopt the assessment approaches from the IPCC's 6th assessment cycle (both SR1.5 and AR6), and to update those methods and assessments in a way that is as closely aligned with those as possible. The attribution assessment here remains very close to that aim, and any differences are detailed in the supplement.

It is not the purpose of this work to re-review the methods themselves (or attribution more generally), and/or to develop a new assessment approach in light of that – this would be the remit of the 7th cycle, or at least a separate paper in its own right.

We do agree that there are limitations to the current methods that could be addressed, and opportunities to improve the current assessment approach, though we disagree about the extent of some of the limitations presented in this review.

*They all use global mean temperatures, GWI - monthly or annual means, KCC - a mixture of multidecadal and annual means, ROF - five year means. This means statistical over/under fitting is of concern and interpreting similarities and differences between the results of the different methods challenging.*

*The GWI method uses a simple model but with no attempt to assess over/under fitting, which is more prone to end-effects than the authors claim, and so it is difficult to understand whether the fit is just a coincidence.*

We disagree that the GWI is more prone to end effects than we have claimed; internal validation tests for the variance of the historical inter-annual revision as a function of each additional year added across the historical period show that the scale of end effects are as stated in the main text (that a particularly hot (cold) year typically only adds (subtracts) around one additional year of warming at current warming rates). If you have information demonstrating that the end effects are significantly larger than we have found, we would be (genuinely) happy for you to send that to us to strengthen these assessments. Secondly, while regression can result in overfitting, internal testing over the historical period suggests

that the role of overfitting is at minor in recent years (and in particular over the years since which the GWI bas been in use, from around 2015 onwards); this is largely since there is low degeneracy between the profiles/shapes of the forced warming components; potential degeneracy can also be accounted for by carefully choosing the aggregations for the forcing components. Presenting details of such tests is beyond the scope of this paper, and we conduct them in order to check that the methods remain similarly valid as in previous years.

*The ROF method uses an approach that effectively has to trade off between variance and bias in its estimates of the climate noise covariance matrix. In normal uses of the method a residual consistency test is done to see if some measure of over/under fitting is noticeable - although that is not discussed in the manuscript (if it was done).*

*The KCC method assumes that the model responses and observations are interchangable, and uses a residual of model responses subtracted from the observations to deduce noise characteristics, so in effect starts out with an assumed anthropogenic and natural attribution. It also assumes the anthropogenic component of the forced response is just the smoothed part of the historical (anthropogenic and natural) response, thus any multidecadal natural responses present will contaminate the "anthropogenic" response used.*

*Fundamentally the net anthropogenic attribution is less informative than the authors portray. All the methods will end up with attributed anthropogenic trends that track the observed trends, as long as long term trends of natural responses is small (Section 6.1.2 in Allen et al, "Quantifying anthropogenic influence on recent near-surface temperature change", Surv Geophys (2006)*

It is worth noting that it isn't the anthropogenic warming component that is fit against the observed warming (which would clearly result in overfitting and a direct tracking of anthropogenic and observed warming), but rather total forced warming is fit against the observed warming. For naturally forced warming trends, we agree that during periods when such natural contributions are small, anthropogenic warming closely tracks the observed warming trend, though we argue that this is evidence of the methods providing useful information about the relative contributions to warming and the limited role of natural trends during those times. Conversely, if the naturally forced contribution to observed warming becomes large (for example during periods of volcanism), this is resolved by the attribution methods, and during those periods anthropogenic warming does not track the observed trend; this deviation is useful information in many contexts. Further, long-term natural trends are accounted for; e.g. in the case of the GWI method through multiple samplings of internal variability that allow for a certain level of natural long-term drifting. While not perfect (e.g. in extremely unlikely cases where the long-term profile of this drift happens to be partially degenerate with the long-term profile of the anthropogenic warming components), this at least factors some amount of long-term natural trends into the resulting uncertainty range.

*"We are fitting all model-simulated signals to the same data so it is not surprising that, after the fitting is done, the scaled model-simulated responses are brought into agreement with each other.") The wider divergence of attributed GHG and OHF supports that view (Figure S8). Including spatial information, could help in gaining confidence in the results, as discussed in Allen et al. 2006.*

*All statistical model/observation analysis methods have different strengths and weaknesses. It does not mean they are perfect, or that they are not useful. Without giving a more fuller context it is not possible for a reader to interpret the results in an informative way.*

We provide lots of comments on the methods in the supplementary material, much more than given in AR6. Again the purpose is to follow AR6 methods.

*Specific comments*
* * *
*L93 No uncertainties are given. If they were it would challenge the "well above" statement.*

It is clear that this comparison applies to best estimates only. We have changed the text to make this clearer. WE change the uncertainty estimate and add and uncertainty range. We now say "more likely than not".

*L137 Not all the data is available in the zenodo repository. Some links are there but not to all of the datasets e.g., one of the observed GMST datasets, precipitation, sea level, GHG concentrations …*

The repository has been updated to now points to all the datasets used.

*Figure 4 top panel, missing N2O label on y-axis.*

This was cropped off by accident, now corrected. Thank you

*Table 2 Are the values in column "2019 (WG1 for ERF estimates)" correct? They appear identical to the F24 table 2 "2019 emission" column. I tried to find the "WG1" values in AR6 but couldn't find them.*

This is indeed from F24 and not AR6 WG1; thank you for pointing out the error in attribution. We have updated the table and figure caption to include the specific citation to be specific. The AR6 values are generated at
https://github.com/chrisroadmap/ar6/blob/main/notebooks/080_chapter7_AR6_ERF.ipynb, the output is at
https://github.com/chrisroadmap/ar6/blob/main/data_output/emissions_ceds_update_plus_bb.csv and the citeable reference is Smith et al. (2021)

Chris Smith, Piers Forster, Matt Palmer, Bill Collins, Nick Leach, Masa Watanabe, Sophie Berger, Brad Hall, Mark Zelinka, Dan Lunt, Michelle Cain, Glen Harris, & Mark Ringer. (2021). IPCC WGI AR6 Chapter 7 (v.1.0). Zenodo. https://doi.org/10.5281/zenodo.5211358

*L502-513 Given the later deep dive into trying to assign causation to year-to-year changes, it seems curious that changes from year to year in natural forcings are glossed over here.*

We feel this is a different discussion, since while the ERF from solar and volcanic variability are clearly variable (and uncertain), they are very obviously natural forcings without human components. The biomass discussion, as other reviewers have alluded to, is much more difficult to separate into anthropogenic and natural effects and as such requires more careful treatment and therefore more words.

*L512-513 The volcanic forcing is not negative according to figure S2, just less positive than recent years.*

We have clarified this represents the HTHH contribution in isolation, and pointed to the section of supplement.

*Section 7.2 and elsewhere. No uncertainties are given for the 2024 GMST value of 1.52C. F24 give uncertainties of about +/-0.1C (90% range).*

We have added a brief discussion of uncertainty range. WE now say

"At the time, 2023 set a new global annual-mean surface temperature change record, with a best estimate of 1.44 ºC, beating 2016 by 0.16 ºC. 2024 surpassed this, reaching a best estimate of 1.52 +/- 0.13 ºC; 2024, becoming the first calendar year since preindustrial more likely than not exceeding 1.5 ºC (Fig. 7). The assessed uncertainty range is based on that in AR6 WGI (Gulev et al., 2021). All four individual datasets are well inside the range (ranging from 1.46 to 1.56 ºC)."

*L 634-635 Given F24 reported uncertainties on GMST of +/-0.1C, it is not "likely" that 2024 is warmer than 1.5C, but rather "as likely as not" - using IPCC convention.*

We have added more discussion on ranges and agree that "more likely than not" is a better descriptor, so use this. See also comment by Richard Allan

*L641-642 Attribution has not been discussed yet, so a reader might be unaware that the "human-induced warming" estimate has a "likely" uncertainty range of 1.1C-1.7C (Table 6). If that is accounted for then the probability is not "1 chance out of 6", but approaching 1 in 2.*

We add extra sentences for clarification and point forward to Section 8.

"These results show that human induced warming combined with particular modes of natural variability shifts the odds of global surface temperatures passing 1.5 oC, making it more likely. Sect. 8 has a fuller discussion of human induced warming."

*L642-644 and Figure 8 I could not find the exact same "framework" in IPCC AR6 WG1 chap 3. Cross-chapter box 3.1 is close, but that does not mention "anthropogenic global warming levels". I do not follow what is being done here. This really needs to be in a study of its own, to get closer attention.*

We disagree, the same approach was used in the cross chapter box of IPCC and a detailed description of the method is given in the paper and our supplement.

*Figure 9 Giving the observed warming for 2024 on the right hand side would be nice.*

*Table 6 Observed first column for "2010-2019" has different uncertainties to that "Quoted from AR6 Chapter 3 Sect. 3.3.1.1.2 Table 3.1" Observed final column "2024" has no uncertainties. It appears the last two columns are "trend-based" (table S5) but equivalent columns in F23 (table 6) and F24 (table 6) are "single year".*

I believe this comment refers to the 1.06 [0.92 to 1.17] value for observed 2010-2019 warming from AR6. We take the 2010-2019 AR6 observed warming from cross-chapter box 2.3, which lists both GMST (1.06 [0.92 to 1.17]°C) and GSAT (1.06 [0.88 to 1.21]°C). Since the attribution assessment reports all results in GMST in this section for consistency (see Supplement §S8.1, and the Table 6 caption), we quote the AR6 uncertainty range for GMST, not GSAT, results here; this is already mentioned in the table and main text. We have now explicitly pointed to Cross-chapter Box 2.3 Table 1 in the table body itself to make tracing these values easier.

*Figure 10 is very different to figure 8 in F24. Bizarrely the natural changes are smaller than shown in F24. I don't understand how this is the case, especially given the amplification in volcanic forcing (Figure 5).*

We agree that the historical period for natural warming in the GWI results is revised compared to previous years; we also note that the historical period for all anthropogenic components in the GWI are almost indistinguishable from previous years. The natural warming level and rate over the historical period is somewhat suppressed compared to previous years, but remains within its uncertainty range. This is likely due to the interaction between the regression algorithm and the revision to the volcanic ERF timeseries this year. This is not the case with the KCC and ROF, since their forcings do not change year-on-year (their annual updates are based on SSP extensions to CMIP6 simulations). This revision to the GWI historical natural warming doesn't affect the headline results, since for 2024 the natural forced level and rate are almost zero in both cases; the effect on the multi-method mean for the level of natural forced warming is therefore negligible. The purpose of Fig. 8 is

to depict the strong correspondance between the ERF trends and attributed warming trends over the historical periods, which remains unchanged.

*Section 11 "Global Land precipitation" How was the GHCN v4 dataset processed. It is provided as station measurements, and not as a gridded dataset. How this is processed is important.*

Response: We applied the AR6 method to calculate global land precipitation using the GHCN v4 dataset as follows: 1. Unit was changed from 0.1mm to mm and data were filtered depending on QC_FLAG from the original GHCN v4 station data. 2. Station data were used if (1) a year has at least 9 months with data, and (2) in each of the 3 periods (1900-1960, 1961-1990, and 1991-2019), at least 50% of the years have data. 3. Grid the selected stations into 5x5 grid boxes and the area global mean precipitation anomaly is computed with respect to 1991-2020 climatology.

*Figure 12 This looks very different to figure 2.15c in AR6 WG1. The change in climatology period is having a striking impact on the evolution of the datasets that have unchanged versions. This should probably be noted.*

Response: The change in climatology period slightly affects the update of figure. The largest change is mainly associated with the version change and data update. For example, there have been significant changes in CRU TS 4.08 compared to CRU TS 4.04 used in AR6, particularly from 1900 to 1940. Even though the same version of GHCN is used, there are some updates in the data.

Good point, text added on this.

*SI L174 Durack 2025 is not in references, but if it is https://egusphere.copernicus.org/preprints/2025/egusphere-2024-3729/ then that paper is a review of what happened up to CMIP6 and does not describe what is planned for CMIP7.*

The Durack reference has been added, it is: Durack, P. J., Naik, V., Nicholls, Z., O'Rourke, E., Turner, B., Buontempo, C., Brookshaw, A., Goddard, C., MacIntosh, C., Hewitt, H., & Dunne, J. P. .: Earth system forcing for CMIP7 and beyond (1.1). Zenodo. https://doi.org/10.5281/zenodo.15469219. 2025.

*SI S5.5.1 and S5.5.2 I don't quite understand the reasoning behind the effort to try to update the volcanic forcing for 2024, but not the solar forcing.*

The solar forcing for CMIP7 covers 1850-2023 with no future (to 2024) extension. This is not yet sufficient to provide an estimate using AR6-like methods with the updated datasets, which requires coverage of pre-1750 to the most recent full year. The CMIP6 time series on the other hand runs from 6500 BCE to 2300 CE, with projections beyond 2014.

*SI L191-208 Giving uncertainties for the reported values here would give helpful context about confidence in the values.*

This is a discussion of different estimate approaches for the recent past where uncertainties are not readily available.

*SI Section S7 the individual observation temperature dataset changes over the periods assessed are not given. This is important for understanding structural uncertainty in the observed trends.*

Individual datasets are discussed in other literature and not the focus of this study or AR6. Replicating the IPCC approach directly requires averaging the datasets. The approach was deliberately conservative. Best guess is mean of all datasets and the range is max/min of all datasets (assuming carry over exactly from AR6). AR6 did not provide this granularity.

*SI L322 - 384 "2023 and 2024 temperature anomalies" This analysis needs to be in its own paper. Given the lack of update to solar forcing and the effort to update the volcanic forcing in section S5.5 it seems odd that here different estimates are used and relied on. In the main text (L487) it is stated that it is "problematic" to attribute year-to-year trends to aerosol forcing. How does that sit with what is being described here? Also I believe it uses a different observed temperature than elsewhere in this paper.*

This section is an illustrative discussion that does not affect the overall conclusion, we want to retain it as it is a useful context to the 2023-2024 warming.

*SI L322 - 384 "2023 and 2024 temperature anomalies" This analysis looks familiar. Might it be worth mentioning it has been published before (albeit it not in a peer reviewed journal). https://wmo.int/publication-series/state-of-global-climate-2024*

Reference added.

*SI L389 "combine measurements of near-surface temperature over land and in some cases over ice" There were no measurements of air temperature over ice, except in some very rare weather ship observations in Arctic circle.*

"Over ice" deleted

*SI L448 "some CMIP6 models may have unrealistically high decadal variability" ... and some will have low decadal variability! As we can't unambiguously observe it, we can't assess which models have realistic decadal variability or not.*

This is s general point, literature indicates that some over estimate is likely in some models.

*SI S8.1 There are inconsistencies here about the importance of differences between GMST and GSAT with what is said in S8.2.1 and S8.2.2. It also rather simplifies the complexities of*

*trying to account for the difference in models and observational datasets, especially over sea-ice when coupled models differ considerably in sea-ice coverage, and there are very few direct measurements of temperature.*

Without the comment explaining what they find to be inconsistent, it is difficult to directly respond to the perceived issue here, as the treatment of the difference between GSAT and GMST reads quite consistently (to my eye).

*SI L509 It might be worth mentioning that the "7%" increase in the best estimate of anthropogenic warming happened when HadCRUT5 had a ~7% increase in warming over HadCRUT4 for 2010-2019. This supports my earlier argument that net anthropogenic attribution is not as informative as we would like.*

This point was already implicitly made on those lines, but the text has now been amended to make this more clear. On the latter point, that estimates of anthropogenic warming are conditional on observed warming doesn't necessarily imply that the net anthropogenic warming attribution is less informative; indeed, if estimates of the forced warming components didn't include information about realised warming, this could be seen as an omission. Convsersely, as estimates of how much warming has actually occurred improve, this in turn improves estimates of the forced warming contributions to that warming.

*SI Table S4 how are the GSAT and GMST results identical for GWI and KCC. They both use GSAT from coupled models at some point. Section S8.2.1 and S8.2.2 try to claim that GMST and GSAT are effectively the same, despite what is said in Section S8.1 and S8.2.3. Can the authors make up their minds? ;-)*

*SI S8.3 I don't think it is explained how the GMST attributed results are estimated from the GSAT results.*

*SI S8.3 The results are presented against the assessed observed temperature changes, but the analyses were done using HadCRUT5. How much impact does this have?*

Yes - ideally the assessed observed warming and assessed attributed warming could be more closely harmonised in this way. However, for consistency the assessments have been conducted as was done in AR6 (where the attribution also used HadCRUT, instead of the IPCC-style multi-dataset observations). The reason for this is that the attribution methods incorporate uncertainty from observed warming, which in turn requires an ensemble for the observed warming, as provided by the HadCRUT dataset. Such an uncertainty ensemble is not available for the IPCC's compound observed temperature dataset, which makes it unsuitable as-is for use in the attribution assessment. Defining the IPCC-style observed warming dataset such that an ensemble is available would be a useful step forwards in the

next assessment cycle. As for its impact, this is reasonably minor; the HadCRUT5 dataset is global mean is

**Response to public comment PC2 by Richard Allan**

We thank Richard for very helpful suggestions and support. We respond to each below

*1) L546 Although EEI dropped in 2024 from a peak of nearly 2 Wm-2 from mid 2022 to mid 2023, due to greater outgoing longwave radiation during the mature phase of the 2023/2024 El Niño, levels did not fall as low as previous mature El Niño evens such as 2016 or 2010 (Allan & Merchant, 2025 ERL doi:10.1088/1748-9326/adb448; Mauritsen et al. 2025 AGU Adv. doi:10.1029/2024AV001636).*

We do not attempt a complete assessment and this section looks at long-term trends rather than variability, so we don't think adding further details is relevant here.

*2) L541 - the increased EEI corresponds with an accelleration of ocean warming, not merely an increase (e.g. Merchant et al. 2025 ERL doi:10.1088/1748-9326/adaa8a)*

We talk about a rate increase which is more or less the same thing. To obtain a quantified estimate of acceleration of ocean warming, a second derivative in time needs to be done. Also, the recent (eg, past 2 decades) increase we see expressed as an acceleration (hence second derivative) would need to be compared with its respective uncertainties (including those with respect to natural variability as 2 decades are short for addressing the long-term change & acceleration) needs to be set in context with the long-term acceleration (eg, from 1960 onwards). This is what has been done in Minière et al., 2023 (paper cited in the manuscript), and the results show that yes, acceleration is larger as compared to the long-term acceleration, but the uncertainties are too large to state that the acceleration observed over the recent 2 decades is larger then the long-term acceleration. The provided reference has been added according to the reviewers suggestion, but wording remains as a rate increase..

*3) L551 - We also showed the EEI increase to originate from increases in absorbed sunlight associated with clouds over the ocean, with hotspots over Californian and Namibian stratocumulus zones (Allan & Merchant, 2025).*

Thank you very much for the suggested reference, which we have included.

*4) L635 - is this an IPCC likely exceeded 1.5oC? A recent WMO synthesis stated 1.55 ± 0.13 based on the 90% confidence range mostly relating to the 1981-2020 minus 1850-1900 IPCC offset applied. I think the 90% confidence range covers 1.645 standard deviations and the 66% level just below 1 standard deviation so this may also span below*

*1.5°C. "Likely" is probably fine but "More likely than not" (>50%) may be more correct. An uncertainty could be shown in Fig.7 or at least mentioned in the caption.*

We have added more discussion on ranges. You are correct so thank you. We agree that more likely than not is more appropriate, so have used this language in our revision.

*5) L651 - the probability of the large jump in global temperatures was increased by the fact the El Niño followed an extended La Niña 2020-2022 (Raghuraman et al. 2024 ACP doi:10.5194/acp-24-11275-2024)*

Statement added inline with comment, using your helpful language - thank you.

*6) L677 - the ocean surface warming is only reconcilable with the raised EEI if the mixed layer shallowed or was compounded by heat from the deeper ocean sequestered during the extended La Niña (Allan & Merchant, 2025 ERL). Shallowing of the mixed layer was liklely to have played a role in the Atlantic (England et al. 2025 Nature in press, published June 19th) while cooling in the 100-300m ocean layer implied a reversal of the downward heat flux below the mixed layer (Minobe et al. 2025 doi:10.1038/s41612-025-00996-z; Allan & Merchant, 2025). It is also likely that a smaller proportion of the EEI directly heated the ocean as more energy was used in heating the atmosphere and probably the land surface (Allan & Merchant 2025 ERL; Minobe et al. 2025).*

We thank the reviewer for the comment, and the provision of further references. Indeed, evidence for the role of heat redistribution in the ocean responsible for change in EEI increases, as particularly outlined int his paper. However, there is still research needed to reconcile these regional processes to change in the global EEI, which are not addressed by this study.

*7) L678 - see also England et al. (2025) Nature (in press) regarding the North Atlantic warming (due out 19 June)*

*Thank you very much, and see comment above - and we fully agree that further analysis is needed to link the regional (large-scale) processes in the ocean to global change in EEI.*

*8) L687 - also Allan & Merchant (2025) ERL relating to the mid 2022 to mid 2023 energy imbalance and the reduced absorption of sunlight over the cloudy ocean*

Reference added

*9) L689 - for global temperatures to stabilise, the peak EEI and decades of decline are required (Mauritsen et al. 2025 AGU Advances, doi:10.1029/2024AV001636, Fig. 2) emphasising the importance of tracking this quantity and serious implications of the risk to the relevant observing systems*

The importance of tracking EEI is discussed elsewhere in the paper.

*10) L1787 - repeated authors from von Schuckmenn et al. 2023 reference*

Thank you, text deleted.

*11) Table3 - I was initially confused by the solar ERF estimate in Figure 5a being for a single year verses the whole solar cycle estimate used and assumed in Table 3. Perhaps a separate symbol could be used for the full solar cycle estimate (and updated to 2014-2024 or corrected for the solar cycle)? However, the single year value is useful since it does contribute a small amount to the larger EEI in 2024 (e.g. Hansen et al. 2025 SPSD doi:10.1080/00139157.2025.2434494; Merchant et al. 2025 ERL doi:10.1088/1748-9326/adaa8a).*

We think the caption of table 5 already covers this detail sufficiently.

*12) Section 8.2 - a couple of studies have noted that the level of annual warmth and the sustained warmth over many months makes it inevitabole that the 1.5 degree Celsius above pre-industrial thershold will be breached (Cannon 2025 Nature Clim. 10.1038/s41558-025-02247-8; Bevacqua et al. 2025 Nature Clim. doi:10.1038/s41558-025-02246-9) without a drastic increase in mitigation (or a large explosive volcanic eruption) so this could be mentioned.*

We don't want to cover projections here, so these studies are not cited.

*13) Section 8.3 could also consider Samset et al. (2023) Comm. Earth Env. doi:10.1038/s43247-023-01061-4 and a preprint by Rahmstorf & Foster doi:10.21203/rs.3.rs-6079807/v1*

We think these studies are less relevant to this analysis which is trying to update AR6 and not reassess.

*14) Section 1 final paragraph - a bulleted or numbered list may work better for the reader to point to the many sections*

We prefer the full paragraph style as written.

**Response to CC3: ['Comment on essd-2025-250'](), Tugberk Samur, 24 May 2025**

*When I check the four observational datasets, I get 1.262 Celsius 10 years' mean anomaly instead of 1.244 for 2015-2024. The RCB comparison between both is huge. If we rely on Zeke Hausfather's estimate for 2025 (1.52), at the end of 2025, that ten years mean will be 1.304. That means we might deplete RCB for 1.5 Celsius (50% likelihood) already in 2025.*
*Based on your dataset, for the next years' paper, you will still mention that we will have some RCB (around 40) while we will be in minus. I think this is a highly sensitive matter because policy-makers take your results into account (UNEP uses this study series for its Emission gap report).*
*I think more discussion in this aspect is critical. We are racing with the time and facing record temperatures which change a lot of assumptions about climate science.*

> We have carefully checked the numbers several times and 1.244 C is correct for the the decadal average of *2015-2024.* GMST numbers are on the github page and Zenodo references. A explicit file of decadal averages is given. WE have checked that this is computed correctly.

**Response to RC2: ['Comment on essd-2025-250'](), Anonymous Referee #2, 28 May 2025**

*The manuscript "Indicators of Global Climate Change 2023: annual update of key indicators of the state of the climate system and human influence" by Forster et al. is a regular update of an extremely thorough assessment of the current state of the anthropogenic climate forcing of Earth and the associated global warming, compiling the most comprehensive state-of-the-art sets of observations and models. An annually updated reference like this is immensely useful in the interim between IPCC assessment reports. This updated reference paper will be essential for any evidence-based discussion and planning related to mitigation of climate change as well as to adaptation.*

*In my opinion, this paper is acceptable as is, bar a few very minor typographical issues listed below and under the assumtion that comments by other reviewers/commenters are dealt with adequately. I have found no further issues which should delay the publication of the manuscript. I do second the comment by Reviewer 1 that the authors should clarify the role of this publication relative to the other two annual update reports mentioned.*

Thank you for your support, we respond to your minor points below

*Minor points:*

*l 279 ... AR6, ...*

*Comma added*

*Fig. 4: The rightmost y axis needs a label (N2O (ppb))*
Label added

*l 352 "of in" should be resolved*
Typo corrected

*Fig. 6a: The two darkest blue colours are almost indistinguishable. Please adjust.*

*Fig. 6b: The stacked bars for the IPCC periods do not work well, particularly in print. I suggest to have the pink and the blue bars beside each*

*others, possibly in half the width each.*

WE have checked print and online versions with colourblind filters and figure looks ok with colours and bars well resolved, so don't quite see the difficulty alluded to. The stacked bars may appear to not work well, because the bars are nearly identical - which is the point we are trying to make with this figure (to reassure the reader that our methods are highly consistent with IPCC AR6).

*l 1085 ...over the course…*
"of " added

*Tab. 11 precip entry: ...due to El Nino…*
Thank you, typo corrected.